# Resolution of the curse of dimensionality in single-cell RNA sequencing data analysis

Yusuke Imoto[1,*] , Tomonori Nakamura[1,2,3,*] , Emerson G Escolar[4,5], Michio Yoshiwaki[5] , Yoji Kojima[1,2,6], Yukihiro Yabuta[1,2], Yoshitaka Katou[2], Takuya Yamamoto[1,5,6] , Yasuaki Hiraoka[1,5,7] , Mitinori Saitou[1,2,6]

**Single-cell RNA sequencing (scRNA-seq) can determine gene expression in numerous individual cells simultaneously, promoting progress in the biomedical sciences. However, scRNA-seq data are high-dimensional with substantial technical noise, including dropouts. During analysis of scRNA-seq data, such noise engenders a statistical problem known as the curse of dimensionality (COD). Based on high-dimensional statistics, we herein formulate a noise reduction method, RECODE (resolution of the curse of dimensionality), for high-dimensional data with random sampling noise. We show that RECODE consistently resolves COD in relevant scRNA-seq data with unique molecular identifiers. RECODE does not involve dimension reduction and recovers expression values for all genes, including lowly expressed genes, realizing precise delineation of cell fate transitions and identification of rare cells with all gene information. Compared with representative imputation methods, RECODE employs different principles and exhibits superior overall performance in cell-clustering, expression value recovery, and single-cell–level analysis. The RECODE algorithm is parameter-free, data-driven, deterministic, and high-speed, and its applicability can be predicted based on the variance normalization performance. We propose RECODE as a powerful strategy for preprocessing noisy high-dimensional data.**

## Introduction

Single-cell RNA sequencing (scRNA-seq) enables the determination of gene expression profiles in multiple individual cells simultaneously (Tang et al, 2009). When used together with recently developed microfluidics platforms and cell index strategies, scRNA-seq permits the analysis of gene expression in thousands of single cells in parallel, accelerating progress in the biomedical sciences (Regev et al, 2017; Cao et al, 2020). On the other hand, because of

technical limitations, scRNA-seq detects only a fraction of the transcriptome in single cells (~1–60%; on average: ~ <10%), and there are large variations in the detection level of each transcript by scRNA-seq (Grun et al, 2014; Kiselev et al, 2019). Thus, unlike conventional bulk RNA-sequencing, scRNA-seq, which is generally used to process numerous cells on automated platforms, provides a sparse representation of the true transcriptome of single cells, with detection failures (dropouts) and variations occurring randomly for most genes, particularly genes with low-expression levels (Lähnemann et al, 2020). These drawbacks, collectively regarded as nonbiological technical noise, pose a key challenge in scRNA-seq data analysis and interpretation.

To circumvent these drawbacks, preprocessing of scRNA-seq data, such as dimension reduction and normalization, is widely used (Stegle et al, 2015; Kiselev et al, 2019). However, such preprocessing does not provide a fundamental solution, because it compresses the original data without separating the true information from the noise information and therefore cannot recover the true expression values. With a focus on mitigating the dropout effects and data sparsity and based on models for transcript/noise distributions as well as on the information from similar cells ("nearest neighbors"), many inventive methods for modifying scRNA-seq data have been proposed (Bonnefoy et al, 2018; Li & Li, 2018; van Dijk et al, 2018; Zappia et al, 2018; Eraslan et al, 2019; Peng et al, 2019; Wagner et al, 2019 Preprint; Wang et al, 2019). These "imputation" methods are classified into several categories, including model-based imputation, data smoothing, and data reconstruction, and appear to be successful in recovering dropped-out gene information (Lähnemann et al, 2020). However, compared with no-imputation controls, most of these methods fail to substantially improve performance in downstream data analyses, such as clustering analyses and dimension reduction mappings; they also introduce "circularity," thereby generating false positives and decreasing the reproducibility of specific gene expressions (Andrews & Hemberg, 2018; Hou et al, 2020). Thus, imputations need to be used with appropriate caution and require further improvements.

---

[1]Institute for the Advanced Study of Human Biology, Kyoto University Institute for Advanced Study, Kyoto University, Kyoto, Japan   [2]Department of Anatomy and Cell Biology, Graduate School of Medicine, Kyoto University, Kyoto, Japan   [3]The Hakubi Center for Advanced Research, Kyoto University, Kyoto, Japan   [4]Graduate School of Human Development and Environment, Kobe University, Kobe, Japan   [5]Center for Advanced Intelligence Project, RIKEN, Tokyo, Japan   [6]Center for iPS Cell Research and Application, Kyoto University, Kyoto, Japan   [7]Center for Advanced Study, Kyoto University Institute for Advanced Study, Kyoto University, Kyoto, Japan

Correspondence: hiraoka.yasuaki.6z@kyoto-u.ac.jp; saitou@anat2.med.kyoto-u.ac.jp
*Yusuke Imoto and Tomonori Nakamura contributed equally to this work.

It is fundamental to note that scRNA-seq data are high-dimensional data (the dimension corresponds to the number of genes, i.e., > ~10,000), with each feature (each gene expression level) bearing technical noise (Grun et al, 2014). High-dimensional statistics theories demonstrate that such data, even when the noises are small, suffer from the "curse of dimensionality" (COD), which causes detrimental effects in downstream data analyses (Hall et al, 2005). Specifically, COD causes impairments of close distances in true data structures, inconsistency of statistics such as contribution rates of principal components (PCs), and inconsistency of PCs, among other deleterious influences (Yata & Aoshima, 2009; Aoshima et al, 2018) (see the Results section and Supplemental Data 1 for details; COD in this context is distinct from that in informatics, which refers to an exponential increase of computational complexity associated with a rise in data dimension). However, despite its significance, COD has not been explicitly addressed in the context of scRNA-seq data analysis, including imputations.

Here, based on high-dimensional statistics (Yata & Aoshima, 2010; 2012), we formulate a noise reduction method, RECODE (resolution of the curse of dimensionality), which resolves COD in scRNA-seq data. RECODE is tailored to the scRNA-seq data with unique molecular identifiers (UMIs) because its mathematical formulation relies on the theory of random samplings, which are involved as noise in the copying and sequencing steps for generating scRNA-seq data with UMIs. We show that RECODE consistently resolves COD in the relevant scRNA-seq data. Significantly, RECODE does not involve dimension reduction, such as selection of highly variable genes (HVGs) and major principal components, for the downstream data analysis, but recovers expression values, even for lowly expressed genes, and thus enables the use of all the gene information and the distinguishing of close cell types/transient cell populations masked by COD. RECODE outperforms representative imputation methods, not only in the cluster level analysis but also in the expression value recovery and single-cell level analysis (e.g., the identification of rare cell types). The algorithm of RECODE is parameter-free, data-driven, deterministic, and high-speed, making the method practical, and notably, the applicability of RECODE is predictable. We propose the use of RECODE as a powerful strategy for preprocessing noisy high-dimensional data, including scRNA-seq data.

# Results

## COD

The noise (technical noise) of scRNA-seq data arises from variations of the copying and sequencing errors during data creation and differs from so-called biological noise, such as transcriptional stochasticity and biological variations. Because true expression values do not contain technical noise, we define the noise of scRNA-seq data as the difference between the observed UMI and true RNA counts divided by their total counts (see RECODE in the Materials and Methods section). A typical cell expresses more than 10,000 genes, and the noise arises in all expressed genes. Accordingly, the accumulation of such noise causes severe problems

in downstream data analyses, so-called COD. In this study, therefore, we attempt to resolve COD in scRNA-seq data analysis.

First, we demonstrate COD using scRNA-seq simulation data with 1,000 cells and variable dimensions (200–20,000) based on the Splatter algorithm (Zappia et al, 2017) (Figs 1A and S1A; see Simulation data creation in the Materials and Methods section). Because the Euclidean distance contains a summation of the squared components, the distance errors of observed values grow according to the dimension. Eventually, the distance errors obscure the difference among neighboring samples. For example, the higher the dimension is, the longer the "legs," that is, distances among neighbor cells/clusters, of the dendrogram of unsupervised hierarchical clustering (UHC) become, leading to an impaired clustering (Fig 1B). Similar problems occur even for other metrics, such as correlation distance (Section 1.1 in the Supplemental Data 1) (Hall et al, 2005). Thus, conventional data analysis methods based on distances fail to identify true data structures in high-dimensional data with noise. We call this type of COD the *loss of closeness* (COD1), and it makes the detailed classification of high-dimensional data impossible.

The noise accumulation also causes adverse effects on data statistics, such as the contribution rate in principal component analysis (PCA) and the Silhouette score (Fig 1C and D). In particular, it has been proven that data variances of PCA-transformed data (eigenvalues of the data covariance matrix) may not converge to true variances for high-dimensional data with noise and low sample sizes (Section 1.2 in the Supplemental Data 1) (Yata & Aoshima, 2009). We call this COD the *inconsistency of statistics* (COD2), and it leads to false statistical inferences.

When there is a considerable variation in the noise scale in each feature, as in the case of scRNA-seq data, another type of COD that induces false PCA structures may occur (Fig 1E). The PCA is robust for noise with small variances, even for high-dimensional data (Section 3 in the Supplemental Data 1). However, in the case of random sampling with a low detection rate, because the variances of some noises become large, the PCA structures will be broken (Yata & Aoshima, 2009). For example, we know that the PCA structures change according to dimension and that the principal components are affected by nonbiological information such as the sequencing depth (the total UMI counts per cell) or the number of detected genes (Fig 1E, Section 4 in the Supplemental Data 1) (Kiselev et al, 2019). We call this type of COD the *inconsistency of principal components* (COD3).

We also note that CODs adversely affect almost all high-dimensional data analysis, including nonlinear dimension reduction methods, such as uniform manifold approximation and projection (UMAP) and t-stochastic neighbor embedding (t-SNE) (Fig S1B and C).

## CODs in scRNA-seq data analysis

We show CODs in real scRNA-seq data using public scRNA-seq datasets generated by 10X Cellular Indexing of Transcriptomes and Epitopes by Sequencing (CITE-seq) data for peripheral blood mononuclear cells (*human PBMC CITE-seq data*, 10X genomics demo data: 10k PBMCs from a Healthy Donor—Gene Expression with a Panel of TotalSeq-B Antibodies; Fig S2A). These data have a complexity level typical for in vivo human cell diversity, and the

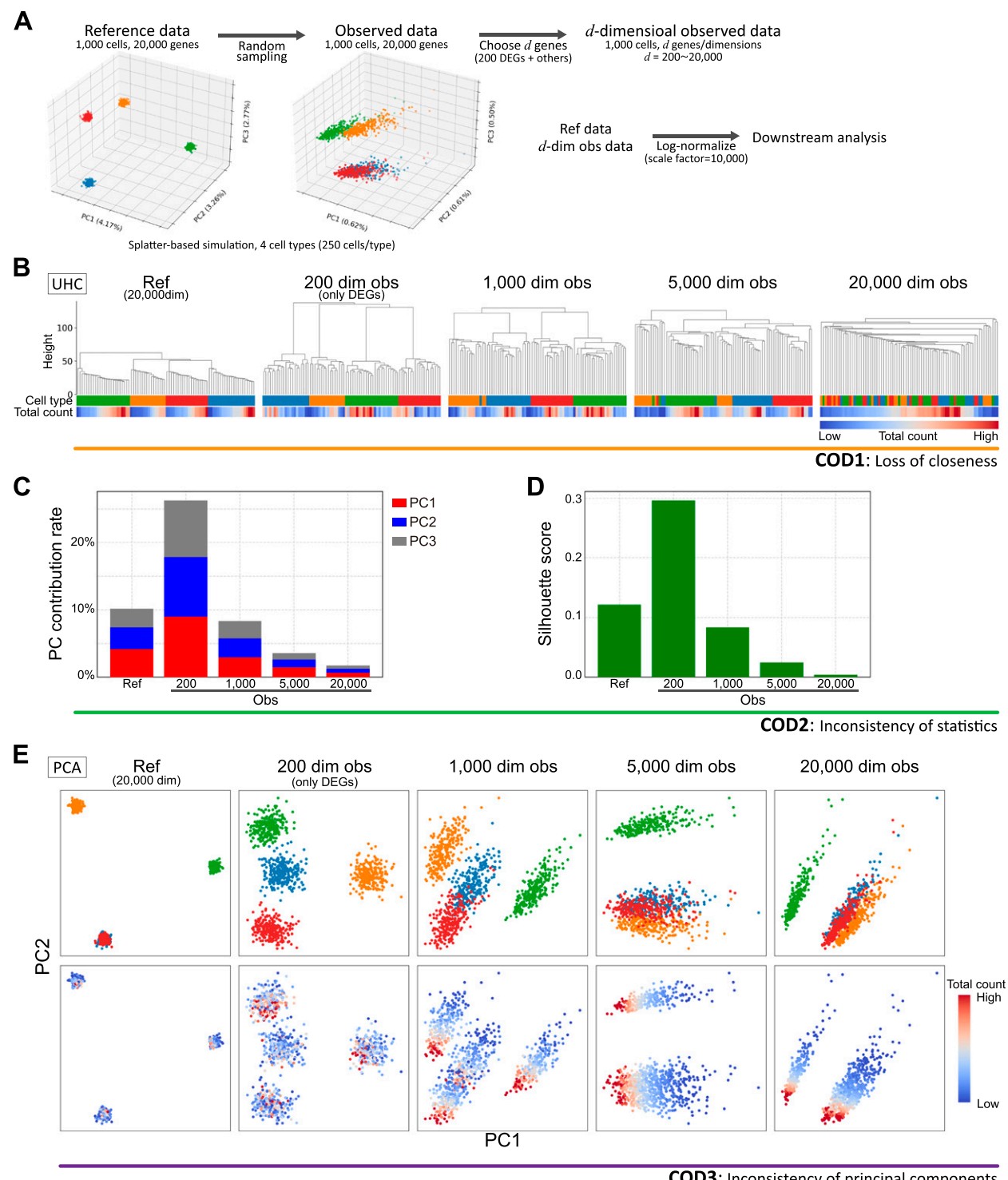

**Figure 1. Demonstration of CODs by scRNA-seq simulation data.**
**(A)** Data creation of reference and observed data by a splatter-based simulation. The simulation scRNA-seq data consists of 1,000 cells, 20,000 genes, and four cell types (250 cells and 50 differential expressed genes [DEGs] per cell type). We created *d*-dimensional observed data by choosing *d* genes so as to contain all DEGs (*d* = 200 (*only DEGs*), 1,000, 5,000, 20,000). We conducted downstream data analyses for the normalized reference and observed data by a log-normalized method that normalizes the expression counts for each cell by the total counts, multiplies this by a scale factor =10,000, and log-transforms the result. **(B)** Demonstration of COD1 (loss of closeness) by an unsupervised hierarchical clustering (UHC) for 100 randomly chosen cells. UHC uses Euclidean distance with cell type and total count labels. The cell type labels get mixed up as the dimension increases. On the other hand, the total count labels are aligned, indicating that the total counts bias the UHC. **(C, D)** Demonstration of COD2 (inconsistency of statistics) by the contribution rates in PCA and the mean Silhouette scores for cell types. **(E)** Demonstration of COD3 (inconsistency of principal components) by PCA projections with colors of the cell types and total counts.

corresponding RNA and protein expression data obtained from the same single cells. We create clusters using the protein expression data of 12 cell surface markers (Figs 2A and S3) and use them as the ground truth clusters. We define the dimension (200, 2,000, and 33,254 [all] genes) by picking up HVGs in order (see Analysis of scRNA-seq data in the Materials and Methods section for details). We demonstrate the CODs using this scRNA-seq data by changing the dimensions.

In the UHC, the legs of the dendrogram become longer as the dimension increases because of COD1 (Fig 2B), leading to the loss of detailed hierarchies. As a result, we may not identify clusters that consist of a small number of cells and thus overlook rare cell types. The contribution rates in PCA and the Silhouette scores become worse (lower rates and lower scores) because of COD2 (Fig 2C and D). Accordingly, we may miss some critical information from the statistical analysis. In the PCA dimension reductions, the major axes (first and second principal components) change as the dimension increases because of COD3 (Fig 2E). In particular, we often observe that clusters grow in one direction, simply following the total UMI counts in scRNA-seq data analysis (Fig 2E bottom); such unidirectional growth arises because of COD3. Therefore, this implies that COD3 affects even PCA processing.

Moreover, CODs have adverse effects on almost all downstream data analyses. For example, UMAP and t-SNE cannot function as the dimensions increase (Fig 2F and G). Using other 10X Chromium and Drop-seq data (10X 3′ scRNA-seq data for five human lung adenocarcinoma cell lines used in a benchmarking experiment [*Cell-Bench data*] [Tian et al, 2019], 10X 3′ scRNA-seq data for human primordial germ cell-like cell [hPGCLC] specification [*hPGCLC induction data*] [Chen et al, 2019], 10X 3′ scRNA-seq data for a mixture of human induced pluripotent stem cells [hiPSCs] and for hPGCLCs [*hiPSC/hPGCLC mixture data*], and Drop-seq data for a cultured cell line [*Drop-seq data*] [Torre et al, 2018] see Figs S2B–E, S4, and S5 and see ScRNA-seq data quality check and preprocessing in the Materials and Methods section for a detailed description of these datasets]), we obtain similar results (Fig S6). In widely used scRNA-seq data analysis tools, for example, Seurat (Satija et al, 2015; Hao et al, 2021) and Scanpy (Wolf et al, 2018), HVGs and/or major PCs are selected for downstream data analyses. However, we do not know a priori the best HVGs to select, and PCA itself is affected by CODs, as we see above.

## Resolution of the curse of dimensionality (RECODE)

To overcome CODs, we herein formulate a novel noise reduction method, RECODE (resolution of the curse of dimensionality). RECODE is tailored to the scRNA-seq data with UMIs because its mathematical formulation relies on the theory of random sampling, which models the copying and sequencing steps for generating scRNA-seq data with UMIs. See RECODE in the Materials and Methods section and Sections 3 and 4 in the Supplemental Data 1 for the detailed mathematical formulations and theorems for RECODE. RECODE consists of four procedures (Fig 3A). Procedure I normalizes the original data by the *noise variance–stabilizing normalization* (NVSN) that transforms the original data such that the noise variances of all features are equal to one. Procedure II projects the normalized data into the PCA space. Procedure III modifies the variances of principal components (eigenvalues of the covariance matrix) based on high-dimensional statistics theories

(*PC variance modification*) for the major PCs (essential part) and by setting those variances to be zero (*PC variance elimination*) for the others (noise part). Procedure IV maps the variance-modified data into the original space by the inverse transformations of I and II. The PC variance elimination and PC variance modification contribute to the resolution of CODs 1–2, respectively, whereas the NVSN resolves the COD3 caused by random samplings.

We show the verification of RECODE using the scRNA-seq simulation data in Fig 1. RECODE shortens the legs of the dendrogram in UHC compared with those in the observed data and, at the same time, achieves more correct clustering (Fig 3B). Furthermore, RECODE obtains better statistics (i.e., better contribution rates in PCA and better Silhouette scores) (Fig 3C and D) and improves the principal components independently of the total counts (Fig 3E). These results indicate that RECODE successfully resolves CODs 1–3. Moreover, the RECODE-preprocessed data are well consistent with the reference data (Figs 3F and G and S7A and B). As a result, downstream data analysis methods, even nonlinear dimension reduction methods, work well even when using all genes (Fig S7C and D). We note apparent overperformances by RECODE in some outcomes, which might stem from the incompleteness of the current scRNA-seq simulation algorithm; see below for RECODE in real scRNA-seq data analysis.

RECODE has the distinctive characteristics of being parameter-free, data-driven, deterministic (no random effect), and high-speed, making it a practical method (Section 5.2 in the Supplemental Data 1). Notably, the applicability of RECODE to scRNA-seq data is predictable based on the variances after the NVSN (hereinafter NVSN variances) (Fig 4A). We can define three classes of the applicability of RECODE to scRNA-seq data. Class A consists of data with NVSN variances of all genes ≥1 and those of many genes = 1; RECODE is applicable to these data with a good noise reduction effect (*strongly applicable*). Class B consists of data with NVSN variances of all genes ≥1 and few/no genes = 1; RECODE is applicable to these data with a limited noise reduction effect (*weakly applicable*). Class C consists of the data not assigned to classes A and B; RECODE is *inapplicable* in these cases. We examined the applicability of the human cell atlas data (*HCA data*) generated by 10X Chromium, Drop-seq, Quartz-seq, Smart-seq2, and Smart-seq3 (Hagemann-Jensen et al, 2020; Mereu et al, 2020; Fig S8). We classified 10X Chromium, Drop-seq, and Quartz-seq data into class A and Smart-seq2 and Smart-seq3 into class B (Fig 4B). We were able to observe the significant reduction of the overvalued variations caused by noise for strongly applicable data (Fig 4C and D). In addition, we note that all 10X Chromium and Drop-seq data we have examined so far belonged to class A. In contrast, there is still overvalued variation in nonsignificant genes in class B, indicating that such data contain an additional noise(s) different from the random sampling noise.

## Resolution of CODs in scRNA-seq data analysis by RECODE

This section verifies the performance of RECODE using relevant scRNA-seq data. We first applied RECODE to the human PBMC CITE-seq data shown in the section "CODs in scRNA-seq data analysis." We show that RECODE resolves CODs 1–3. In the UHC (Fig 5A), we can observe the closer distances (shorter legs) among neighbor cells

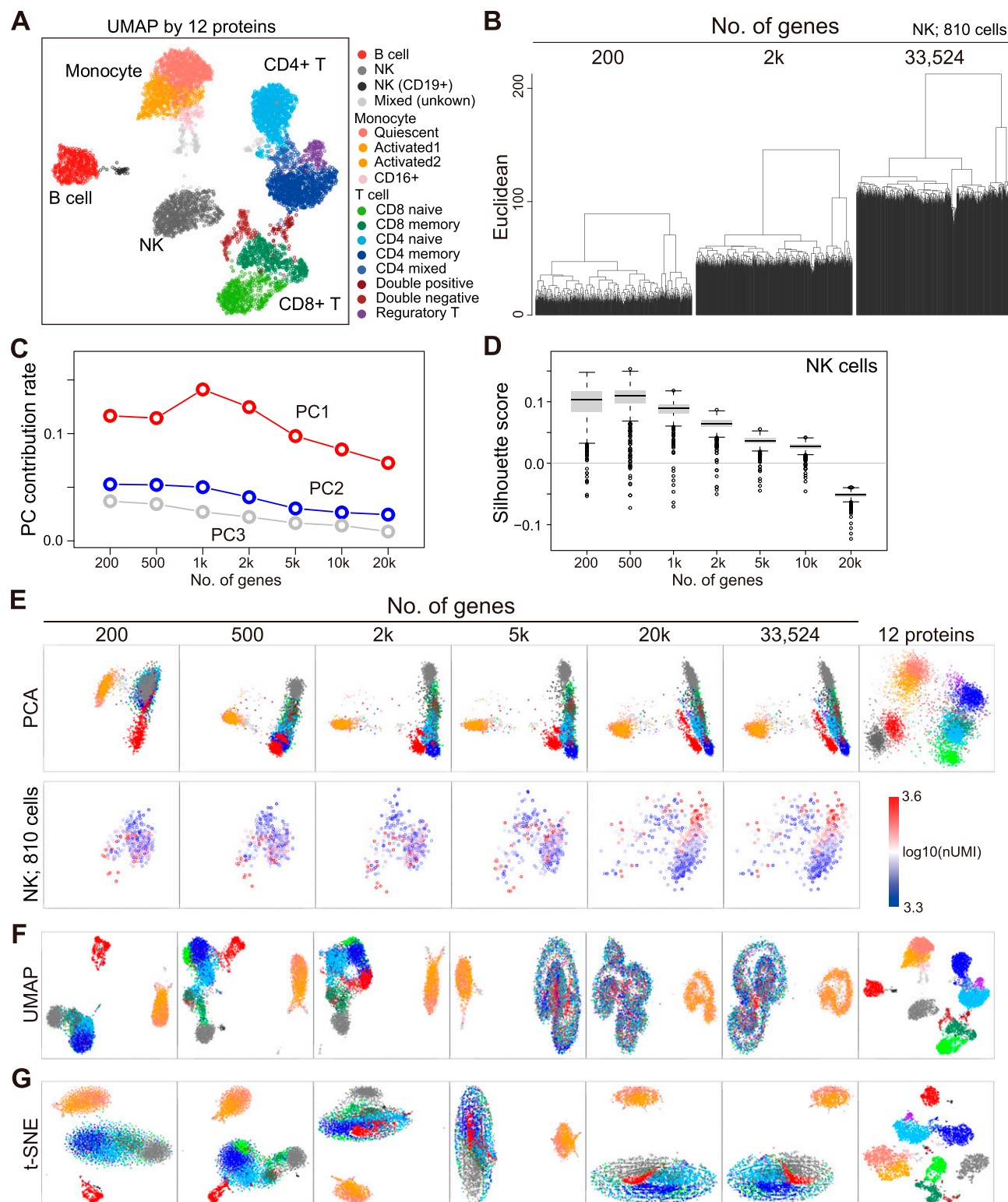

**Figure 2. Demonstration of CODs in real scRNA-seq data (human PBMC CITE-seq data).**
**(A)** UMAP plots computed by 12 protein expressions [$\log_2$(ss.median + 1)]. The cells are colored by the cell annotation defined in Fig S3. **(B)** Demonstration of COD1 by clustering with 200, 2,000, and 20,000 (HVGs) for NK cells. HVGs were selected by the FindVariableFeatures function in the Seurat package. **(C)** Demonstration of COD2 by the contribution rates in PCA. **(D)** Demonstration of COD2 by the Silhouette score for the NK cells. **(E)** Demonstration of COD3 by PCA projection of the whole cells colored by the cell types defined in (A) (top) and the NK cells colored by total UMI counts (bottom). **(F, G)** Nonlinear dimension reduction mappings by UMAP (F) and t-SNE (G) colored by the cell types.

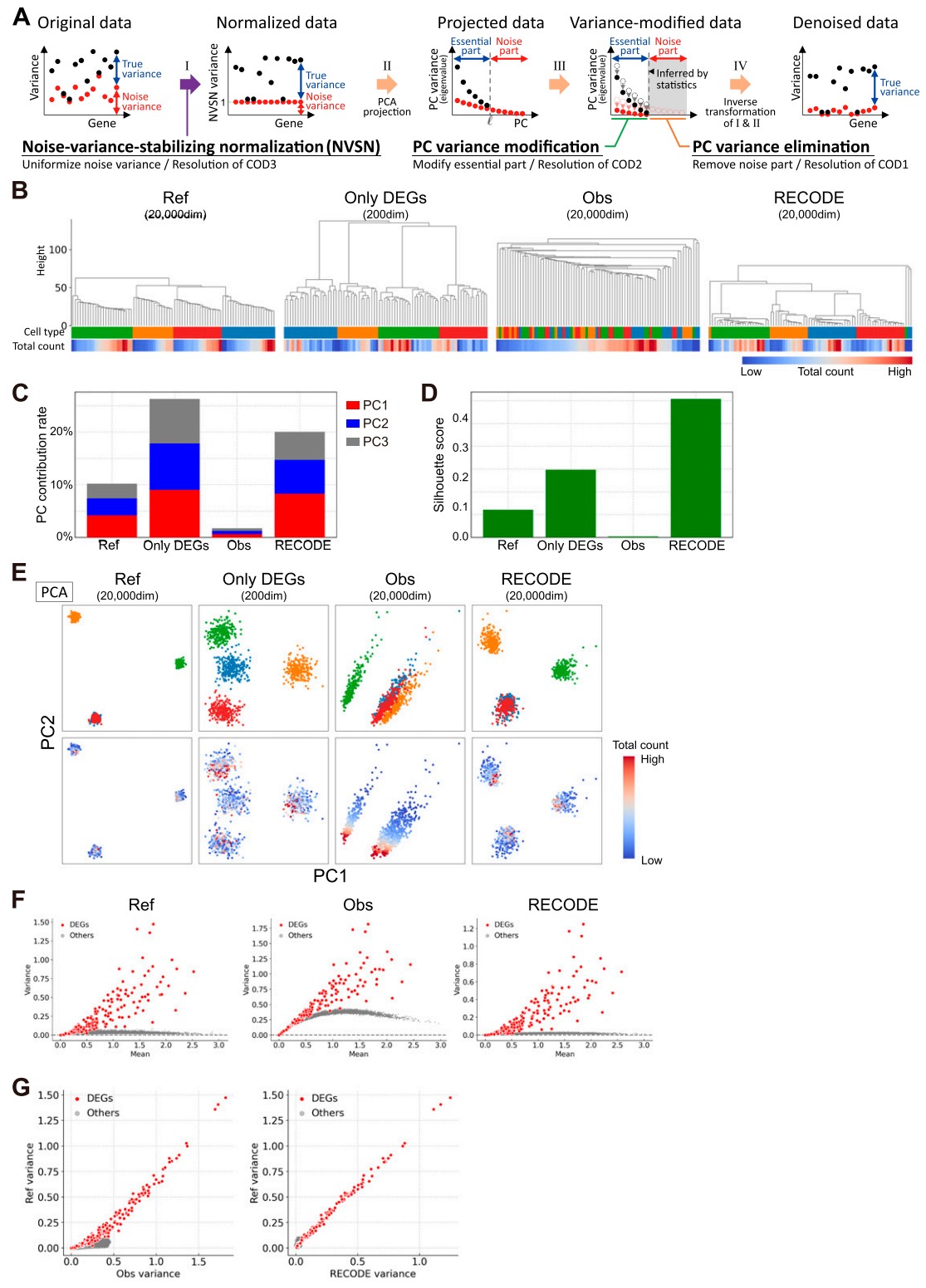

**Figure 3. RECODE algorithm and verification by scRNA-seq simulation data.**
**(A)** Sketch of four procedures in RECODE. The black and red dots show the variances of observed data and noise, respectively, for genes. **(B, C, D, E)** Demonstrations of the resolution of CODs 1–3 by RECODE. **(B)** Dendrogram by UHC using Euclidean distance with cell type and total count labels. **(C)** Contribution rate in PCA. **(D)** Mean Silhouette score for cell types. **(E)** PCA projections with colors of the cell types and total counts. RECODE-preprocessed data show the high identification of cell types and better scores for statistics. **(F, G)** Comparison of variances of genes among reference, observed, and RECODE variances after log normalization by mean versus variance plot (F) and biaxial plots of reference/observed variances and reference/RECODE variances (G). The RECODE variances are highly correlated with the reference variances.

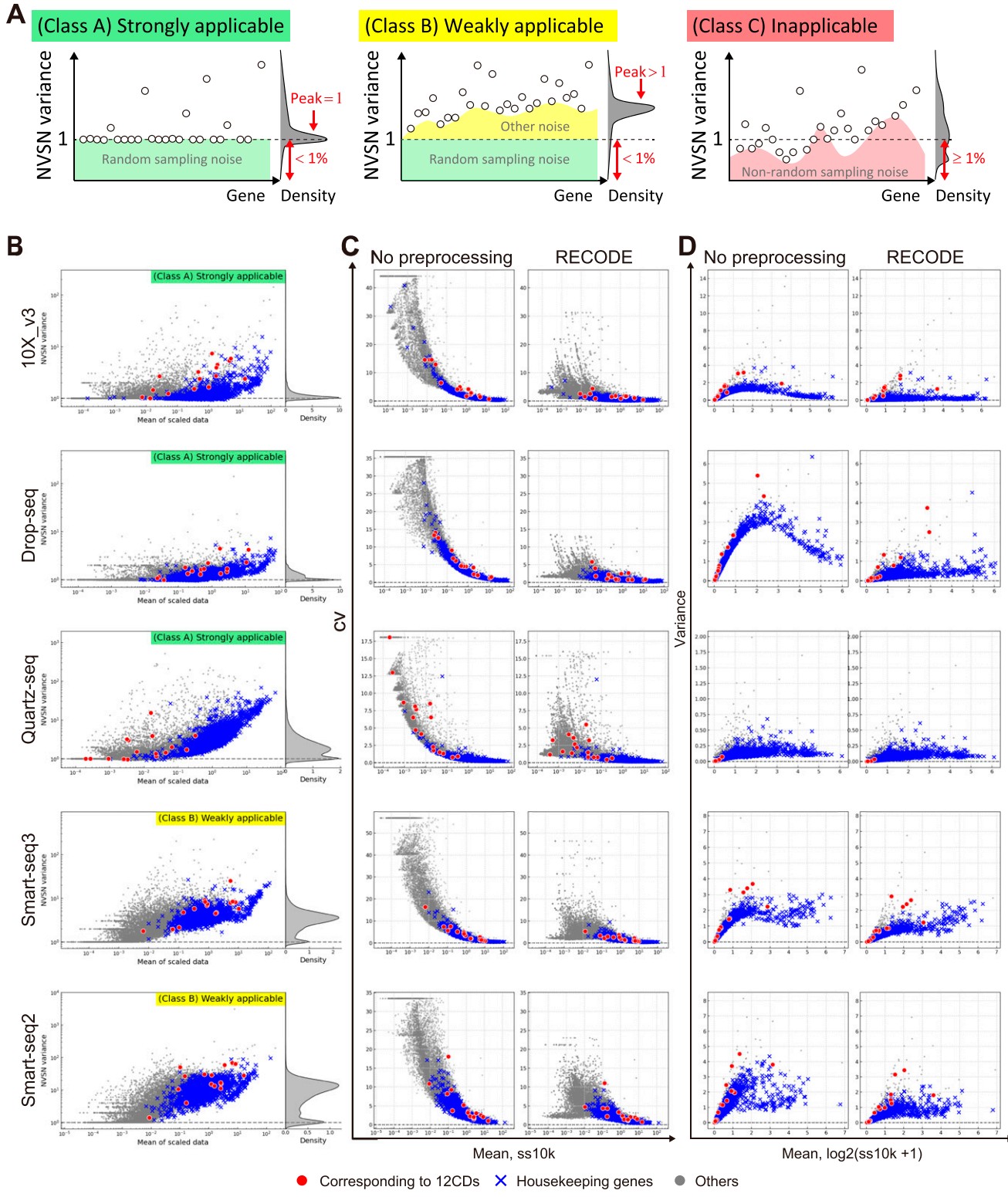

**Figure 4. RECODE applicability for scRNA-seq platforms.**
**(A)** Sketch of the three classifications of RECODE applicability. **(B)** RECODE applicability to the HCA data generated by the library creation platforms, 10X Chromium (version 3), Drop-seq, Quartz-seq, Smart-seq 2, and Smart-seq 3 (Hagemann-Jensen et al, 2020; Mereu et al, 2020). **(C, D)** Comparisons of coefficients of variation and variances with no preprocessing (before RECODE) and RECODE-preprocessed data. The markers in (B, C, D) show the CD genes corresponding to the observed proteins in CITE-seq (red circles), housekeeping genes (blue crosses), and the other genes (gray circles).

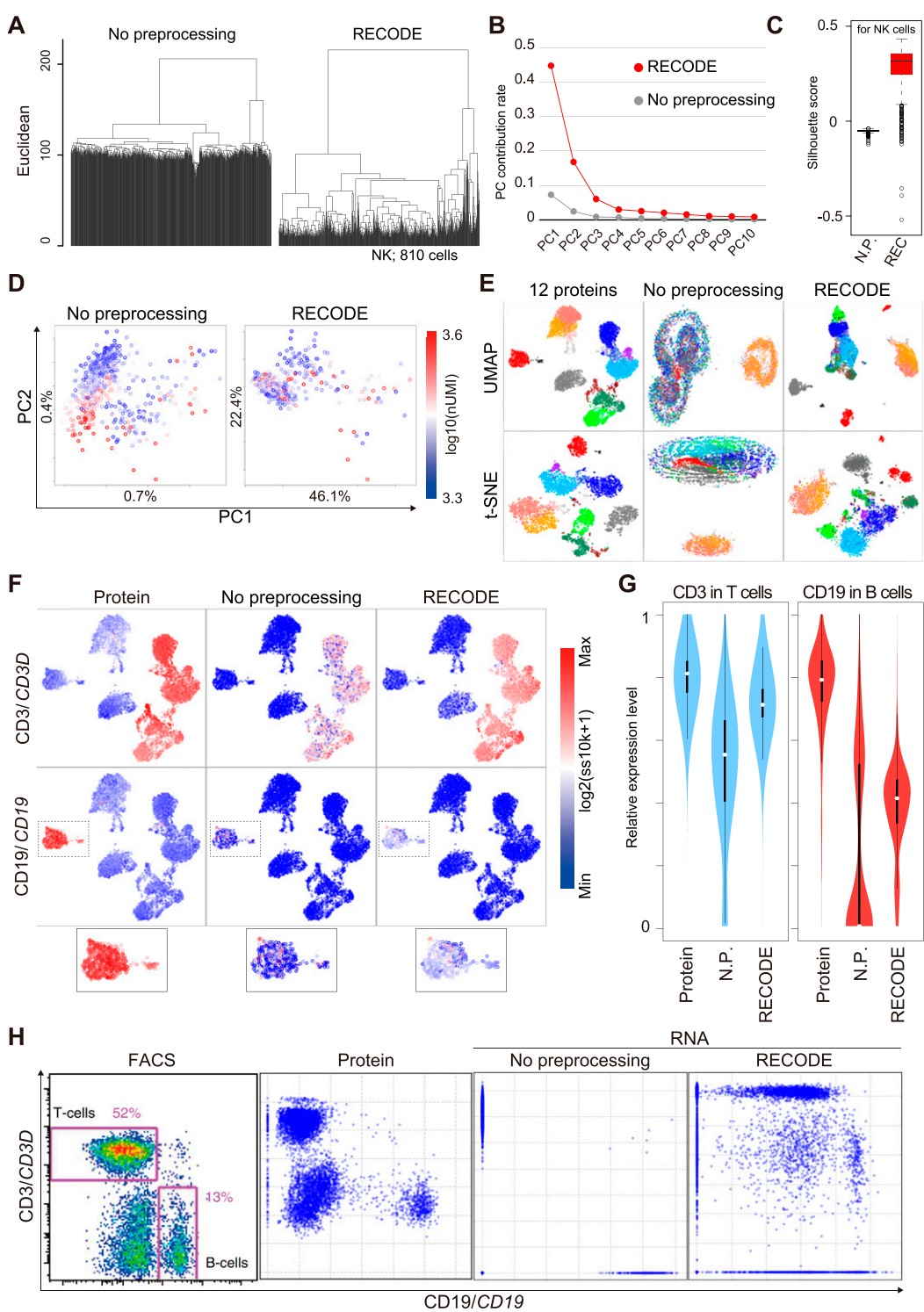

**Figure 5. Verification of RECODE in real scRNA-seq data.**
**(A, B, C, D)** Demonstration of the resolution of CODs 1–3 by RECODE. **(A)** Dendrogram by UHC for the NK cells. **(B)** Contribution rate in PCA of the human PBMC CITE-seq data. **(C)** Silhouette score for the NK cells. **(D)** PCA projections of the NK cells colored by total UMI counts. **(E)** Nonlinear dimension reduction mappings by UMAP (upper) and t-SNE (bottom) for the human PBMC CITE-seq data using 12 protein expression data (see Figs 2A and S3) and all gene expression data (33,524 genes) with/without RECODE. The colors indicate the cell types. **(F)** UMAP plots of the human PBMC CITE-seq data using 12 protein expressions colored by the expression values of CD3/*CD3D* and CD19/*CD19*. The bottom panels indicate the enlarged insets of the CD19/*CD19*–positive B-cell cluster. **(G)** Violin plots of relative expression levels for CD3/*CD3D* (left) and CD19/*CD19* (right) in T-cell and B-cell clusters. N.P. denotes no preprocessing (without RECODE). The relative expression indicates the expression level when the maximum log$_2$-transformed expression value in the data is set as 1. **(H)** Biaxial plots for CD19/*CD19* and CD3/*CD3D* expressions in the human PBMC CITE-seq data. The leftmost image is the actual FACS plot (Stoeckius et al, 2017).

and more detailed hierarchies after RECODE (resolution of COD1). This enables us to readily classify the populations of fewer cells. In the statistics analyses by the contribution rates of PCA and the Silhouette scores (Figs 5B and C), we obtain better values that enable us to discuss the statistical significance after RECODE (resolution of COD2). In the PCA projection (Fig 5D), RECODE returns the principal components independent of the total number of UMIs (resolution of COD3). As a consequence of resolving CODs, RECODE can improve the mappings of nonlinear dimension reduction methods, such as UMAP and t-SNE (Fig 5E).

Next, we show that RECODE recovers the expression value of each gene. In the UMAP plot colored by the expression values of the lowly expressed genes *CD3D* and *CD19* (Fig 5F), we observe scattered blue dots (zero expression values) in the non-preprocessed data, which are regarded as dropouts. In contrast, almost all the blue dots disappear after RECODE, and the relative expression levels of *CD3D* and *CD19* get close to those based on the protein expressions (Fig 5F and G). Moreover, in the biaxial plots of *CD3D* and *CD19* (Fig 5H), the distribution of the non-preprocessed data is too sparse and is completely disparate from that of the protein expressions measured by FACS (Stoeckius et al, 2017) and by CITE-seq because of the severe dropout effects. This indicates that the current scRNA-seq data analysis without preprocessing does not capture the true gene-to-gene relations among lowly expressed genes. In contrast, the RECODE-preprocessed data distribution becomes closer to the distributions of FACS and the protein expression. We also obtained similar results with the other datasets. For example, similar findings were obtained using the CellBench data (shorter legs and detailed hierarchies in UHC [Fig S9A], improved PC contribution rates [Fig S9B], improved Silhouette scores [Fig S9C]), the hPGCLC induction data (improved PC contribution rates [Fig S9B], improved mapping in t-SNE and UMAP [Fig S9E and F], recovery of gene expression values [Fig S9G], recovery of gene-to-gene relationship [Fig S9H]), and the hiPSC/hPGCLC mixture data (improved PC contribution rates [Fig S9B], improved Silhouette scores [Fig S9C]), and the Drop-seq data (independence of clustering from total UMI counts [Fig S9D]).

From the verification above, we conclude that RECODE resolves CODs and thus can recover the gene expression values. Thus, RECODE functions effectively as a noise reduction method for scRNA-seq data.

## Comparison of RECODE with imputation methods

We compare the performance of RECODE with that of representative imputation methods using the human PBMC CITE-seq data. We use six imputation methods that show high performance in previous reports (Andrews & Hemberg, 2018; Hou et al, 2020) and employ different imputation categories (Lähnemann et al, 2020)—namely, model-based imputation (SAVAR and scImpute) (Li & Li, 2018; Huang et al, 2018), data smoothing (DrImpute and MAGIC) (Gong et al, 2018; van Dijk et al, 2018), and data reconstruction/matrix factorization (ALRA and ENHANCE) (Wagner et al, 2019 *Preprint*; Linderman et al, 2022). We do not use machine learning–based methods because their results are strongly dependent on the training data or hyperparameters.

We first investigate the performance upon UMAP without any pre-dimension reduction (e.g., PCA) (Fig 6A). ScImpute and DrImpute

failed to map cells in an appropriate manner, for example, NK cells (colored in gray) were intermingled with other cell types. Furthermore, we found that ENHANCE recognized a number of scRNA-seq data as identical data (43 out of 6,341 [~0.7%]), presumably because of its aggregation algorithm, leading to the loss of a fraction of single-cell information. The Silhouette scores with RECODE, SAVER, MAGIC, ENHANCE, and ALRA, but not with scImpute and DrImpute, are improved over those with no preprocessing (Fig 6B). The scores with MAGIC and ENHANCE are even higher than those of the other methods, which might also be because of their aggregation algorithm (see below). Next, we show the performance on the variance correction (Fig 6C). scImpute, DrImpute, and ALRA could not reduce the variances of nonsignificant genes (e.g., housekeeping genes) appropriately, whereas RECODE, SAVER, MAGIC, and ENHANCE suitably reduced them while preserving those of significant genes. We conclude that the four methods, scImpute, DrImpute, ALRA, and ENHANCE, bear flaws as a single-cell noise reduction methodology.

Next, we investigate the performance of RECODE, SAVER, and MAGIC on the recovery of gene-to-gene relationship by examining the *CD3D* and *CD19* expression values preprocessed with the three methods. By coloring the expression values on the UMAP plot (Fig 6D), we find that all the methods recover their expression to a level comparable to that detected by CITE-seq protein expressions. Furthermore, the relative expression levels also become significant (Fig 6E). These two results show similar performance of RECODE, SAVER, and MAGIC in the cluster level analysis. In the biaxial plots for *CD3D*/*CD19* (Fig 6F), which can be regarded as a single-cell–level analysis, three major populations (*CD3D*$^+$/*CD19*$^-$: T cells; *CD3D*$^-$/*CD19*$^+$: B cells; and *CD3D*$^-$/*CD19*$^-$: the other cell types) are found in the FACS/CITE-seq protein expression plots. We compare these plots with the scRNA-seq data without preprocessing and with preprocessing by RECODE, SAVER, and MAGIC. Without preprocessing, the plot patterns are very different from those of the FACS/CITE-seq protein expression plots, and the frequencies of the *CD3D*$^+$/*CD19*$^-$ and *CD3D*$^-$/*CD19*$^+$ populations are lower, whereas that of the *CD3D*$^-$/*CD19*$^-$ population is higher than those of the FACS/CITE-seq protein expression plots (Fig 6F). In contrast, preprocessing by RECODE, SAVER, and MAGIC all recover the abundance ratios of the *CD3D*$^+$/*CD19*$^-$ and *CD3D*$^-$/*CD19*$^+$ populations. On the other hand, we observe the aggregated distributions with MAGIC compared with the FACS/CITE-seq protein expression and the sparse and discrete distribution in low expression levels with SAVER. In contrast, like FACS/CITE-seq protein expression, the biaxial plot of RECODE shows a continuous distribution and the three major populations.

To explore this point further, we next applied RECODE, SAVER, and MAGIC to the Drop-seq data of a cultured cell line, for which RNA-FISH data of 26 genes are also available as a ground truth (Torre et al, 2018). Biaxial plots of four pairs of genes show that in contrast to the non-preprocessed data, the RECODE-preprocessed data show a distribution highly similar to that of the RNA-FISH data (Fig 7A). In contrast, unlike the RECODE-preprocessed data, both the SAVER- and MAGIC-preprocessed data show somewhat over-aggregated distributions (Fig 7A). To gain quantitative insight into this aspect of the analysis, we computed relative errors from the second- to sixth-order moments of their distributions (the second-order moment is the variance), which revealed lower error rates by RECODE than by SAVER and MAGIC in most

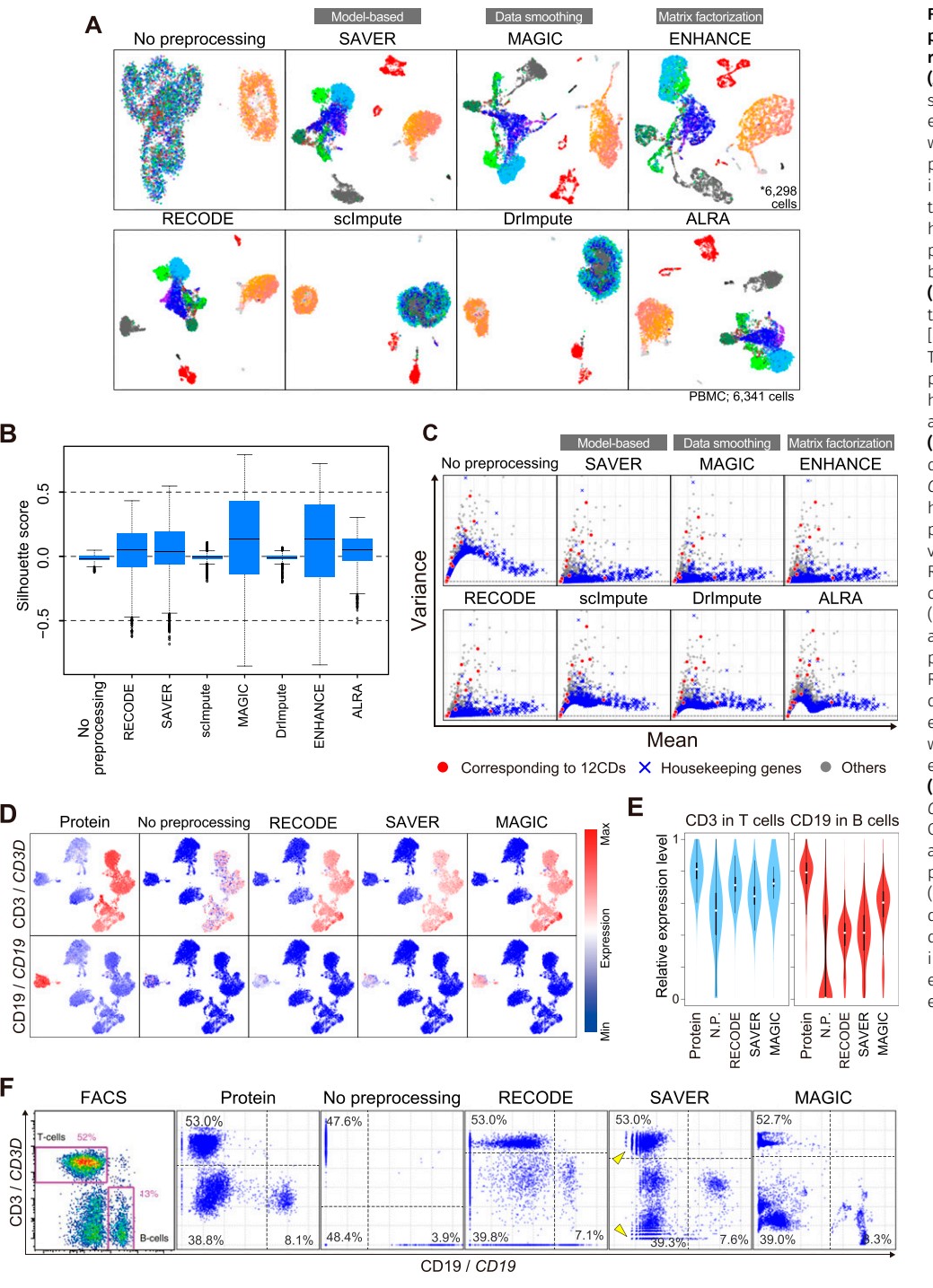

**Figure 6. Comparison of the performance of RECODE and representative imputation methods.**
**(A)** UMAP plots of the human PBMC CITE-seq data with a full set of RNA expression profiles (33,254 genes) without preprocessing and with preprocessing by RECODE and imputation methods. **(B)** The boxplots of the Silhouette scores for all the cells in human PBMC CITE-seq data without preprocessing and with preprocessing by RECODE and imputation methods. **(C)** Scatter plots of the mean versus the variance of gene expression [log₂(ss10k+1)] for the same data as in (A). The genes corresponding to the 12 proteins are colored in red, the housekeeping genes (Hounkpe et al, 2021) are in blue, and the others are in gray. **(D)** Protein and RNA expression distribution, CD3/*CD3D* (top) and CD19/*CD19* (bottom), on UMAP plots of the human PBMC CITE-seq data using the 12 protein expressions. The RNA expression values were preprocessed with RECODE, SAVER, or MAGIC. **(E)** Violin plots of relative expression levels for CD3/*CD3D* (left) and CD19/*CD19* (right) in T-cell and B-cell clusters without preprocessing and with preprocessing by RECODE, SAVER, and MAGIC. N.P. denotes no preprocessing. The relative expression indicates the expression level when the maximum log₂-transformed expression value in the data is set as 1. **(F)** Biaxial plots for CD19/*CD19* and CD3/*CD3D* expressions in the human PBMC CITE-seq data without preprocessing and with RECODE, SAVER, and MAGIC preprocessing and the actual FACS plot (Stoeckius et al, 2017). The percentages denote the frequency of cells in each quadlet (population). Arrowheads indicate the sparse and discrete expression value distributions in low expression levels introduced by SAVER.

of the moments for these genes (Fig 7B). These findings unequivocally demonstrate that the quantitative performance of RECODE is better than those of SAVER and MAGIC.

To further compare the performance of RECODE, SAVER, and MAGIC in the single-cell resolution analysis, including the detection of rare cell populations, we designed mixed scRNA-seq data that contain major and rare cell types using CellBench data and hiPSC/hPGCLC mixture data. We generated pseudo-rare cell types with the variable number of cells (100, 20, 10, 5, 3, 1 cells) (Fig 8A and B). We applied RECODE, SAVER, and MAGIC to the mixed scRNA-seq data and monitored gene expression values signifying rare cell types (*ZBED2* and *CA9* for CellBench data and *OTX2* and *ZIC3* for hiPSC/hPGCLC mixture data) (Fig 8C and D). In the cases of SAVAR and MAGIC, the expression levels were negatively influenced by those of the major cell type. In addition, as reported in a previous study (Tian et al, 2019) and shown in Figs 6F and 7, SAVER sometimes returns

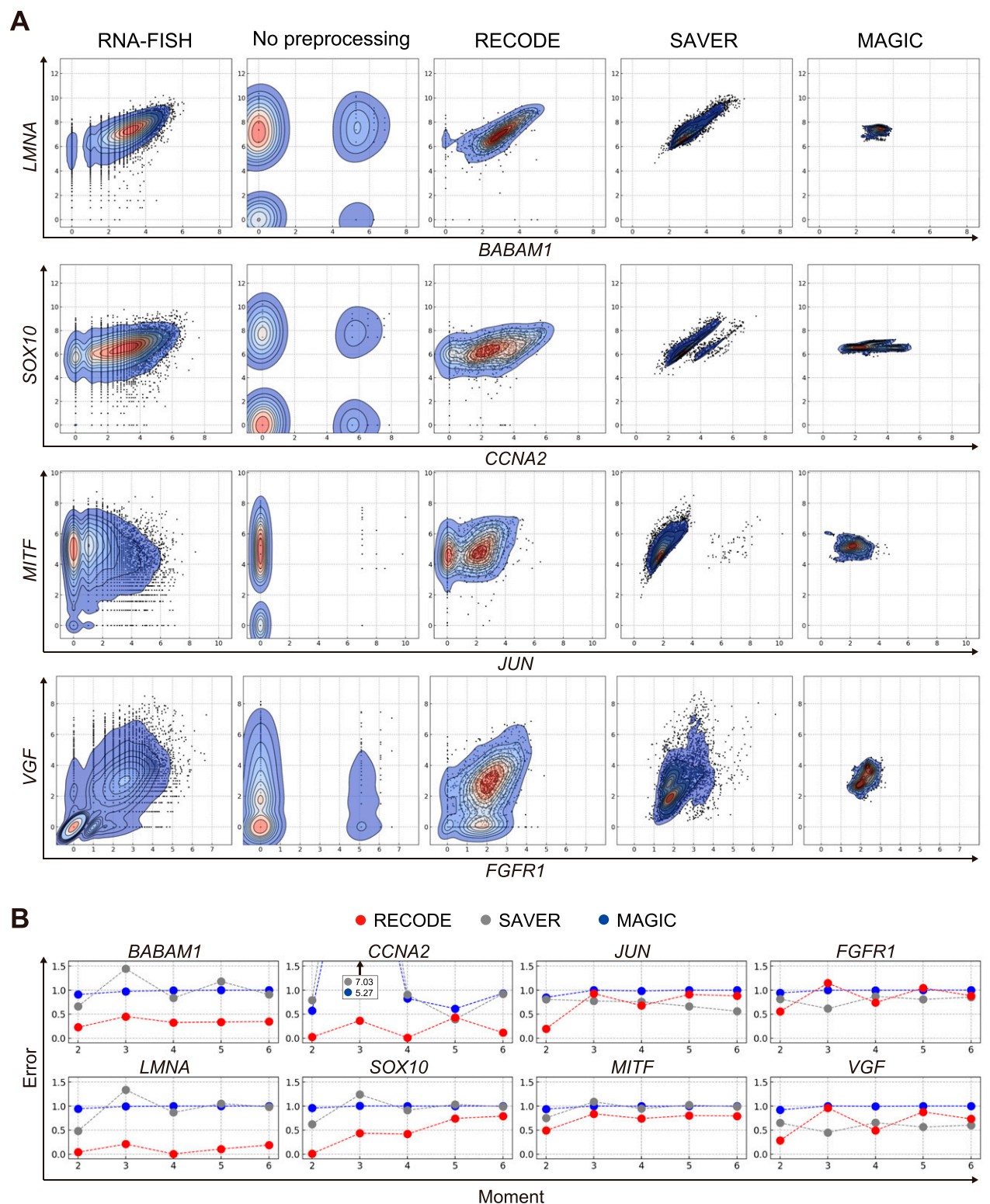

**Figure 7. Comparison of the performance of RECODE, SAVER, and MAGIC for recovery of the gene-to-gene relationship based on RNA-FISH data.**
**(A)** Biaxial plots for gene expressions (*BABAM1/LMNA*, *CCNA2/SOX10*, *JUN/MITF*, and *FGFR1/VGF*) in the single-molecule FISH (Torre et al, 2018) and Drop-seq data without preprocessing and with preprocessing by RECODE, SAVER, and MAGIC. Both axes are log-scaled. **(B)** Relative errors from the second- to sixth-order moments (the second-order moment is the variance) of the distributions for the genes in (A). The distributions of RECODE are more similar to RNA-FISH data and more accurate than those of SAVER and MAGIC.

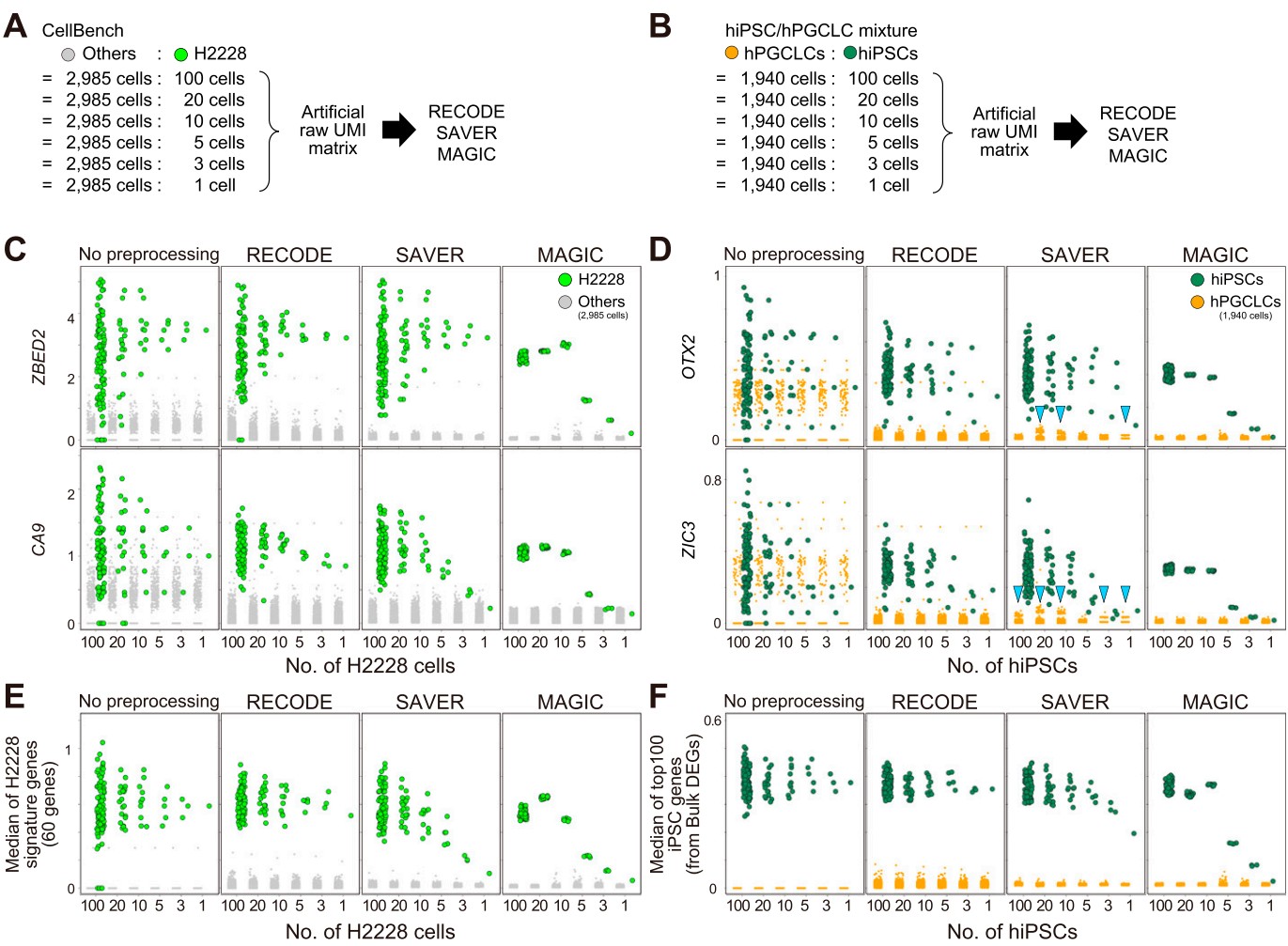

**Figure 8. Comparison of the performance of RECODE, SAVER, and MAGIC for rare cell-type detection.**
**(A, B)** Schematic views of artificial raw unique molecular identifier matrix creation. Set 100, 20, 10, 5, 3, and 1 cell(s) of the pseudo-rare cell types (H2228 and hiPSCs) and merge them with the other cell types (2,985 other cells in CellBench data and 1,940 hPGCLCs in hiPSC/hPGCLC mixture data). **(C, D)** The expression values of highly expressed genes (*ZBED2/CA9* in CellBench data (C) and *OTX2/ZIC3* in hiPSC/hPGCLC mixture data (D)) for pseudo-rare and other cell types. The columns show the expressions without preprocessing and with preprocessing by RECODE, SAVER, and MAGIC. The arrowheads indicate the artifacts introduced by SAVER. **(E, F)** The median expression of highly expressed genes, defined by bulk RNA-seq data analysis (see Figs S7 and S8), for pseudo-rare and other cell types.

discrete values (arrowheads in Fig 8D). In contrast, RECODE does not lose essential signals even with one cell. Moreover, we observe this trend even in the median of signature genes (Fig 8E and F). Consequently, only RECODE can identify a true heterogeneous gene expression at the single-cell resolution and detect rare cell types hidden by noise.

In addition, we compared the computational practicalities of RECODE, SAVER, and MAGIC. The run-time of RECODE is as fast as that of MAGIC, which is one of the fastest imputation methods (Hou et al, 2020) and much faster than that of SAVER (Fig 9A). The high-speed performance of RECODE is achieved by a fast algorithm based on mathematical treatments of eigenvalues (Section 5.1 in the Supplemental Data 1). Furthermore, the scalability of RECODE for the number of cells was the best (RECODE:0.91, SAVER:1.12, MAGIC:1.86, calculated by the linear regression of run-times for 10,000–30,000 cells), indicating that RECODE can be applied to a

greater number of cells in the most practical manner. Moreover, the memory usage of RECODE is as low as that of MAGIC (Fig 9B).

## Precise delineation of cell fate transition dynamics and identification of rare cell populations during mouse gastrulation by RECODE

This section explores the performance of RECODE for the delineation of cell fate specification dynamics and identification of rare cell populations on a complex dataset for mouse gastrulation from embryonic day (E) 6.5 to E8.5 (*mouse gastrula data*, 10X Chromium, version 1) (Pijuan-Sala et al, 2019). This previous study has identified many cell types in a successful manner, but there are some misannotated or unidentified/undescribed cell populations. For example, the node, an organizer for the left–right axis determination (Lee & Anderson, 2008), was misannotated as the notochord, which

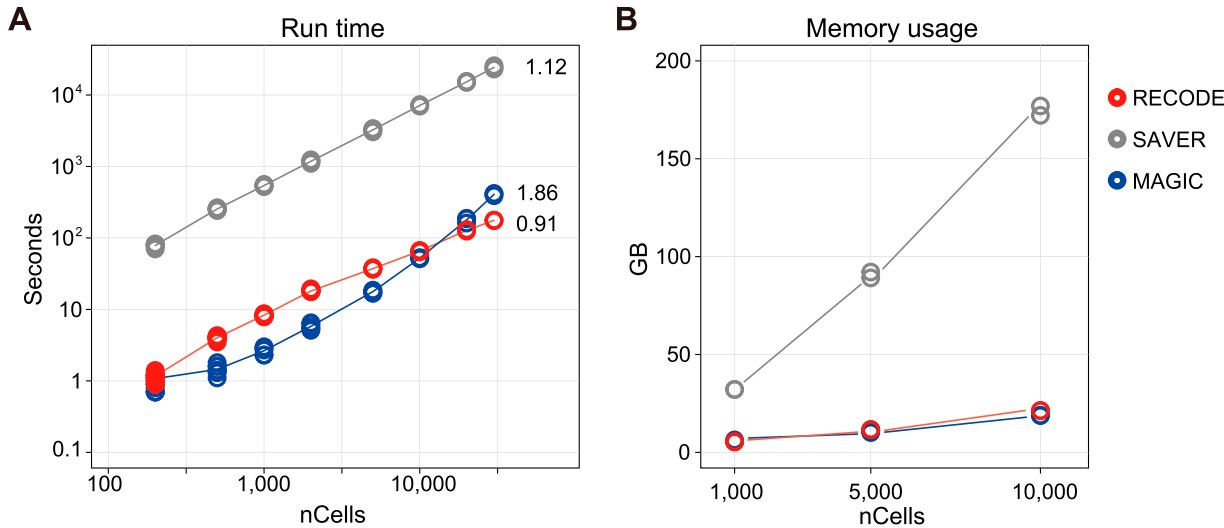

**Figure 9.   Practicality of RECODE.**
**(A, B)** Run-time (A) and memory usage (B) of the RECODE, SAVER, and MAGIC methods for 200–30,000 cells. The cells were picked from the pool of the hPGCLC induction data (see Evaluation of run-time, memory, and scalability in the Materials and Methods section for details). The numbers in (A) indicate the scalabilities of RECODE (0.91), SAVER (1.12), and MAGIC (1.86), which were calculated by the linear regression of run-times for 10,000–30,000 cells.

is a distinct rod-like structure formed both anteriorly and posteriorly to the node and plays a key patterning function, such as in the dorsal–ventral axis patterning of the neural tube (Balmer et al, 2016). Furthermore, anterior visceral endoderm (AVE), an essential signal center for the anterior–posterior axis formation (Bardot & Hadjantonakis, 2020), was also not annotated/identified.

We compare the performance of RECODE with that of preprocessing in Seurat, a widely used scRNA-seq data analysis platform based on dimension reduction (Satija et al, 2015; Hao et al, 2021). The preprocessing in Seurat consists of the selection of HVGs (2,000 genes) and major principal components based on the jackstraw algorithm for the original data (see Analysis of scRNA-seq data in the Materials and Methods section). We conduct the subsequent downstream analyses under the same conditions. First, we analyze the mouse gastrula data for E7.5 embryos, in which complex cell fate specifications/embryonic patterning proceeds, including node and notochordal plate formation, using Seurat and RECODE preprocessing. On UMAP plots (Fig 10A), key lineage-marker gene expressions of the Seurat-preprocessed data capture progressive cell fate transitions from the epiblast (EPI) to mesoderm (Meso) and to extraembryonic mesoderm (Ex.Meso): it appears that $Pou5f1^+$ EPI goes on to express key mesoderm markers such as $T$ and $Hand1$ successively, leading to the formation of $Pou5f1^-/Tal1^+$ Ex.Meso. However, the $Pou5f1^{low}/Sox17^+$ definitive endoderm (D.Endo) is isolated from both the EPI-Meso–Ex.Meso cluster and the $Ttr^+$ VE cluster. In contrast, RECODE-preprocessed results delineate continuous cell fate transitions from $Pou5f1^+$ EPI not only to Meso and Ex.Meso but also to D.Endo through $Pou5f1^+/T^+$ cell populations. Furthermore, D.Endo appears to be extending toward a subpopulation of the VE cluster that expresses $Sox17$ at a weak level. It has been demonstrated that during gastrulation, D.Endo arises from the $T^+$ mesendoderm (Bardot & Hadjantonakis, 2020), D.Endo

ingresses into the VE layer, and both D.Endo and VE contribute to the gut endoderm (Kwon et al, 2008). During this process, D.Endo continues to be $Ttr^-/Sox17^+$, whereas the $Ttr^+/Sox17^-$ VE becomes $Ttr^+/Sox17^+$ cells (Viotti et al, 2014). Thus, these findings indicate that the RECODE-preprocessed data recapitulate continuous cell fate transition dynamics during mouse gastrulation in a more appropriate manner than the Seurat-preprocessed data. This would be because although the Seurat-preprocessed data lose critical information that reflects precise cell-to-cell relationships during dimension reduction (i.e., HVG and PC selection), the RECODE-preprocessed data recover the expression values of all genes, including lowly expressed ones.

Next, we apply UHC to Seurat- and RECODE-preprocessed data to determine a more precise annotation of cell types (Fig S10A–C). Under the UHC results and key gene expressions, we first annotate the TE, VE, and EPI lineages (Fig S10A). Then, we classify the EPI lineage based on the UHC results and the expression of the Meso and D.Endo markers and the genes that are expressed in the node and/or notochordal plate ($T$, $Mesp1$, $Hand1$, $Tal1$, $Sox17$, $Noto$, $Foxj1$, $Pifo$, $Chrd$, and $Cer1$) (Biben et al, 1998; Plouhinec et al, 2004; Yamanaka et al, 2007; Hadjantonakis et al, 2008; Cruz et al, 2010; Kinzel et al, 2010; Balmer et al, 2016) (Fig S10B and C). Accordingly, we identify the node essentially as $Noto^+$, $Foxj1^+$, $Pifo^+$, $Chrd^+$, $T^+$, $Cer1^-$ cells and the notochordal plate essentially as $T^+$, $Cer1^+$ cells in both datasets. We note that the numbers of the cells classified into the respective subclusters differ to a certain extent, and more importantly, the hierarchical position of the node differs between the Seurat- and RECODE-preprocessed data (Fig S10C). Consistent with the UHC result, the node is also isolated on the UMAP plot of the Seurat-preprocessed data (Fig 10B). In contrast, the UMAP plot of the RECODE-preprocessed data shows a continuous transition from the $Pou5f1^+/T^+$ EPI cells to the node, notochordal plate, and D.Endo. This is in good agreement with the fact that the node,

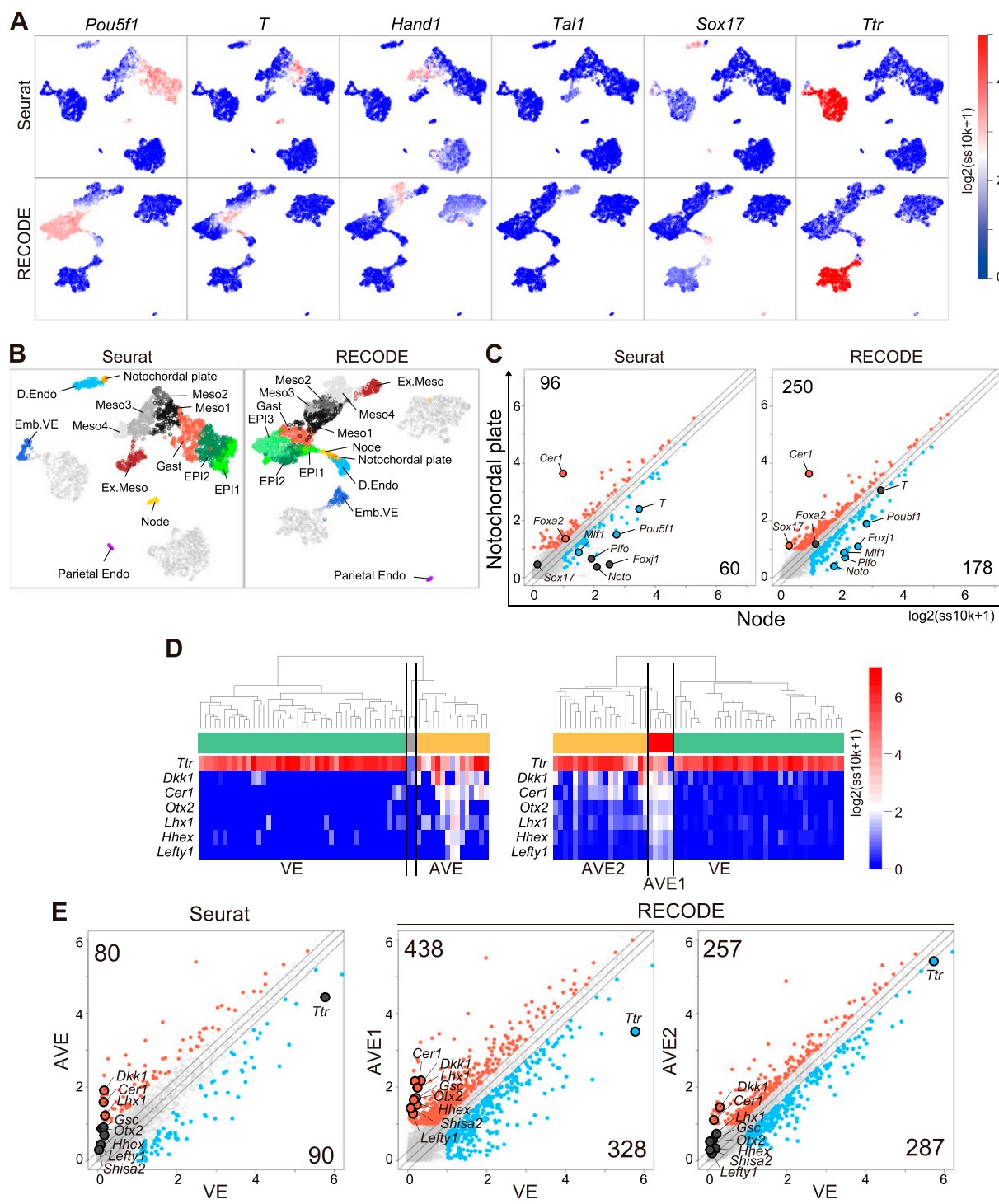

**Figure 10. Precise delineation of cell fate transition dynamics and identification of rare cell populations during mouse gastrulation by RECODE.**
**(A)** UMAP plots of cells consisting of E7.5 mouse embryos with Seurat (top) or RECODE (bottom). The colors show the expression levels of the indicated genes. **(B)** UMAP plots of Seurat and RECODE colored by the clusters defined in Fig S10B and C. **(C)** Scatter plots of the averaged gene expression levels between the node and notochordal plate at E7.5, defined by the unsupervised hierarchical clusteringUHC of Seurat and RECODE in Fig S10C. The DEGs of node (blue) and notochordal plate (red) were defined as > 0.25-fold difference (flanking diagonal lines), mean log₂(ss10k + 1) > 1, and false discovery ratio < 0.05. The other genes are colored in dark gray, the annotated genes show key genes, and the numbers of corners show the numbers of DEGs for node and notochordal plate cells. **(D)** Enlarged UHC dendrogram of the *Ttr*-high VE lineage at E6.5 in Fig S10E and heatmap of the levels of selected markers for AVE with Seurat or RECODE. **(E)** Scatter plots of the averaged gene expression levels between VE and AVE clusters of Seurat (left), VE, and AVE1 clusters of RECODE (middle) and VE and AVE2 clusters of RECODE (right) at E6.5. The DEGs of VE (blue) and the DEGs of AVE, AVE1/AVE2 (red) were defined by the same criteria as in (C). The other genes are colored in dark gray, the annotated genes show key genes, and the numbers of corners show the numbers of DEGs for each cell type.

notochordal plate, and D.Endo are derived differentially from the mesendoderm, that is, the node originates from the anterior end of the primitive streak, the notochordal plate arises from the anterior midline, whereas D.Endo forms around flanking areas (Balmer et al, 2016). Thus, the RECODE-based analysis successfully captures accurate developmental trajectories, which were overlooked in the Seurat-based analysis.

Next, we show differentially expressed genes (DEGs) between the node and the notochordal plate using the Seurat- and RECODE-preprocessed data (Fig 10C). We obtain a substantially larger number of DEGs of the RECODE-preprocessed data than those of the Seurat-preprocessed data for both the node and the notochordal plate (178 and 250 DEGs for RECODE and 60 and 96 DEGs for Seurat). Although we detect *Cer1* as a notochordal plate-specific gene in the Seurat- and RECODE-preprocessed data, we identify *Noto*, *Pifo*, and *Foxj1* as statistically significant, node-specific DEGs, only in the RECODE-preprocessed data (Fig 10C). Moreover, the Seurat-preprocessed data define *T* and *Foxa2* as DEGs specific to the node and notochordal plate, respectively, whereas it has been shown that they are expressed uniformly in the node and notochordal plate (Lolas et al, 2014; Balmer et al, 2016), indicating a false-positive call (Fig S10D).

Next, we analyze the mouse gastrula data for E6.5 embryos. We classify TE, VE, and EPI lineages by the UHC results and key gene expressions (Fig S10E). We further classify the VE by the expression of AVE markers (*Dkk1*, *Cer1*, *Otx2*, *Lhx1*, *Hhex*, *Lefty1*) (Takaoka & Hamada, 2012; Bardot & Hadjantonakis, 2020) (Fig 10D). We find that Seurat-preprocessed data can classify the VE cells into two major clusters (VE and AVE), including one that expresses key AVE markers, but this classification might be uncertain because of the sparse and variable expression values. In contrast, the RECODE-preprocessed result can clearly classify the VE cells. Moreover, the AVE cells can be classified into two cell types, with one expressing AVE markers at higher levels (5 cells: AVE1) and the other expressing them sporadically (19 cells: AVE2). In the DEG analysis (Fig 10E), we obtain larger numbers of DEGs both between AVE1 and VE (438 and 328 DEGs) and between AVE2 and VE (257 and 287 DEGs) with the RECODE-preprocessed data compared with the number of DEGs obtained between AVE and VE (80 and 90 DEGs) with the Seurat-preprocessed data. Importantly, two genes that are known to be specific to AVE, *Gsc* (Belo et al, 1997; Ding et al, 1998) and *Shisa2* (Furushima et al, 2007; Cheng et al, 2019), are contained in only DEGs of the AVE1 cells with the RECODE-preprocessed data. These findings indicate that the AVE1 represents the genuine AVE, and the AVE2 may be the cells located around AVE1 with a property akin to AVE, implying that RECODE can produce further biological insights.

## Discussion

We have formulated a noise reduction method, RECODE, which resolves the COD in scRNA-seq data analysis. RECODE significantly mitigates CODs caused by random sampling noises in creating scRNA-seq data with UMIs. Accordingly, we have shown that RECODE significantly reduces those noises and recovers the gene expression values, even for lowly expressed genes, enabling us to use all of the information of genes in the downstream data analysis

and allowing the discrimination of single cells bearing close gene expression profiles and identification of rare cells. We have performed comprehensive comparisons of RECODE with representative imputation methods, including SAVER and MAGIC, and a widely used scRNA-seq data analysis platform (Seurat), demonstrating the superiority of RECODE. RECODE is parameter-free, data-driven, deterministic, and high-speed, and importantly, the applicability of RECODE can be predicted by variance distributions after noise variance–stabilizing normalization (NVSN).

The successful application of RECODE to scRNA-seq data with UMIs is based on the fact that those noise variances are appropriately estimated from the theories of random sampling in statistics (see RECODE in the Materials and Methods section). The procedures for generating the scRNA-seq data consist of essentially three key steps: copying the original information (reverse transcription of mRNAs), amplifying the copied information (PCR amplification of cDNAs), and sequencing the amplified information (next-generation sequencing of amplified cDNAs) (see Section 4 in the Supplemental Data 1). The copying and sequencing steps are assumed to involve random sampling noise because these steps randomly pick up mRNAs and cDNAs, respectively, whereas the amplifying step involves a distinct type of noise resulting from the PCR amplification, which can vary for each cDNA molecule. The introduction of UMIs in the cDNA synthesis step before the cDNA amplification offsets this noise, allowing an approximation of noises associated with scRNA-seq as random sampling noises. Indeed, RECODE mitigated random sampling noises in all the scRNA-seq data generated on the 10X Chromium and Drop-seq platforms we evaluated, irrespective of their chemistry versions (Fig 4). We found that although Smart-seq3 involves UMIs, it is only weakly applicable to RECODE. In contrast to the typical methods—such as scRNA-seq on the 10X Chromium platform—that incorporate UMIs in the 3′ ends of mRNAs at the beginning of cDNA synthesis, the Smart-seq3 method provides UMIs at the 5′ ends of full-length cDNAs after the cDNA synthesis (Hagemann-Jensen et al, 2020). This would create additional noise(s) because the cDNA synthesis reaction often stops in the middle of mRNAs, and full-length cDNA synthesis depends on various variables, such as mRNA length and nucleotide compositions (Nakamura et al, 2015). We reason that these potential additional sources of noise make Smart-seq3 only weakly applicable to RECODE.

On the other hand, scRNA-seq data based on read counts contain mixed noises (random sampling noise and PCR amplification noise), and the extent of noise reduction by RECODE would not be as large as that for the scRNA-seq data with UMIs (Fig 4). Similarly, merged data among different culture environments or developmental time points include other noises known as batch effects. These data are beyond the scope of the current version of RECODE because their noises are not regarded as those consisting of only random sampling noise. To optimize the noise reduction effect by RECODE on such data, it is imperative to appropriately estimate the noise variances brought about by the mixture of random sampling, PCR amplification, batch effects, and possibly other factors. Toward this end, we need to examine the manner and the mechanism of noise emergence in each step of the scRNA-seq data generation in a more careful fashion and construct a mathematical theory to model the process for such noise emergence.

Although both RECODE and imputation methods directly modify raw data values, they focus on different aspects of data and, accordingly, employ distinct principles. Typically, imputation methods are parametric and aim to circumvent the dropout effect and data sparsity. To recover the expression of dropped-out genes based on their expression levels in nearest neighbors, imputation methods require some prior data clustering (e.g., k-nearest neighbors), which does not escape from COD and therefore potentially leads to incorrect results. Information aggregation of nearest neighbors does not work for rare cells with no biologically appropriate neighbors, leading to the loss of such rare cells (Fig 8). Furthermore, because these methods are not necessarily based on a precise understanding of the noise-generating processes, they may improve a part of the original data but exacerbate other parts, causing problems such as "over-imputation" or "cyclicity" (Andrews & Hemberg, 2018; Lähnemann et al, 2020). In contrast, RECODE is nonparametric and focuses on removing COD based on theories in high-dimensional statistics. Accordingly, RECODE mitigates data sparsity and simultaneously allows a more quantitative comparison of cells bearing close data structures.

With respect to the categories of imputation methods (Lähnemann et al, 2020), RECODE can be categorized as a model-based method because its key underlying theory is the noise variance modeling of random sampling data. At the same time, RECODE uses the PCA (singular value decomposition) and can therefore be categorized as a matrix factorization as well. Thus, RECODE employs multiple theories based on the formulation of scRNA-seq data generation.

Importantly, RECODE does not impose any assumptions on the data types for their application; that is, RECODE is applicable to other sequencing data created by a similar platform. For example, with an appropriate preprocessing, the spatial gene expression and scATAC-seq data with 10X Chromium are also categorized as class A (strongly applicable) (Sections 5.3 and 5.4 in the Supplemental Data 1). In addition, RECODE can be applied to high-dimensional data with noise in biology and even in other disciplines. We would therefore like to propose that RECODE presents a powerful strategy for preprocessing noisy high-dimensional data.

## Materials and Methods

### RECODE

We propose a noise reduction method for scRNA-seq data represented by UMI counts called RECODE (resolution of the curse of dimensionality). Using high-dimensional statistical theories, RECODE can effectively reduce the noise in data through the resolution of the curse of dimensionality. The detailed theories are shown in the Supplemental Data 1.

Let $c_{ij}$ ($i = 1,...,d, j = 1,...,n$) be the observed UMI count of gene $i$ in cell $j$ and $c_{ij}^{\text{true}}$ be its true expression value. Here, $n$ and $d$ are the sample size and dimension, respectively. Because the scale of $c_{ij}$ and

$c_{ij}^{\text{true}}$ can vary significantly because of the low detection rate in data sampling, we consider the following scaled values (probability):

$$x_{ij} = \frac{c_{ij}}{t_j}, \ x_{ij}^{\text{true}} = \frac{c_{ij}^{\text{true}}}{t_j^{\text{true}}},$$

where $t_j$ and $t_j^{\text{true}}$ are the total UMI and RNA counts in cell $j$, that is, $t_j = \sum c_{ij}$ and $t_j^{\text{true}} = \sum c_{ij}^{\text{true}}$, respectively. We model the noise $e_{ij}$ ($i = 1,...,d, j = 1,...,n$) defined as the difference between the observed and true scaled values:

$$e_{ij} := x_{ij} - x_{ij}^{\text{true}}.$$

Hereafter, we treat the above values as random variables.

To correctly split the essential and noise parts in the singular value decomposition-based transformation in RECODE, we first address COD3 (inconsistency of principal components) caused by the nonuniform noise (procedure I in Fig 3A). Based on the random sampling procedures, we model the observed UMI count data $c_{ij}$ as

$$c_{ij} \sim \text{Poisson}\left(t_j x_{ij}^{\text{true}}\right).$$

This model generally coincides with previous studies (Grun et al, 2014; Huang et al, 2018; Hafemeister & Satija, 2019). Some recent studies have also modeled the parameter distribution in the Poisson distribution as a gamma distribution. As a result, the mixture model follows the negative binomial distribution. Although the negative binomial model may represent scRNA-seq data, estimating two independent parameters is a serious problem with high computational costs. In contrast, this study does not assume the distribution of the parameter $t_j$ (allowing any distributions). That is, this study employs a general distribution that contains distributions such as the Poisson, gamma, and negative binomial distributions used in previous studies.

Without modeling the parameter distribution $t_j$, we directly proved the relationship of the following statistics for genes (see Theorem 4.1 in the Supplemental Data 1):

$$\text{Var}_i\left(x_{ij}\right) = \text{Var}_i\left(x_{ij}^{\text{true}}\right) + \text{E}_i\left(x_{ij}/t_j\right). \tag{1}$$

Here, $\text{E}_i$ and $\text{Var}_i$ are the expectation (mean) and variance for fixed gene $i$. To see the behavior of the noise variance, let us set $x_{ij}^{\text{true}}$ to be constant. Then, we obtain $\text{Var}_i(e_{ij}) = \text{E}_i(x_{ij}/t_j)$ from Equation (1). The conventional studies based on the negative binomial distribution have considered that the mean of scaled data describes the noise variance as $\text{Var}_i(e_{ij}) = a\text{E}_i(x_{ij}) + b\text{E}_i(x_{ij})^2$ ($a$, $b$: parameters). In contrast, our observation $\text{Var}_i(e_{ij}) = \text{E}_i(x_{ij}/t_j)$ indicates that the explanatory variable is $\text{E}_i(x_{ij}/t_j)$. This explanation of the noise variance is an essential difference from the conventional studies. Moreover, we can directly evaluate the noise variance from observed data without parameters.

Using a function $f_\alpha$ ($\alpha \geq 0$) given as

$$f_\alpha(x) = \begin{cases} \dfrac{x}{\sqrt{\alpha}}, & \alpha > 0, \\ 0, & \alpha = 0, \end{cases}$$

we define normalized data $z_{ij}$ as $z_{ij} := f_{E_i(x_{ij}/t_j)}(x_{ij})$. We consider the noise of the normalized data as $e'_{ij} := z_{ij} - z_{ij}^{\text{true}}$, where $z_{ij}^{\text{true}} = f_{E_i(x_{ij}/t_j)}(x_{ij}^{\text{true}})$. Because the noise $e'_{ij}$ satisfies $\text{Var}_i(e'_{ij}) = 1$ for expressed gene $i$, we call the transformation $x_{ij} \rightarrow z_{ij}$ the *noise variance–stabilizing normalization* (NVSN).

Next, we denoise normalized UMI count $z_{ij}$ by transforming eigenvalues (procedures II and III in Fig 3A). Let $Z = (z_{ij}) \in \mathbb{R}^{d \times n}$ be the matrix form of normalized data and $\overline{Z} \in \mathbb{R}^{d \times n}$ be the matrix of row averages. We consider the singular value decomposition of centralized data as $Z - \overline{Z} = U_Z \Sigma_Z V_Z$, where $U_Z$, $\Sigma_Z$, and $V_Z$ are a $d \times d$ orthogonal matrix, a $d \times n$ rectangular diagonal matrix, and an $n \times n$ orthogonal matrix, respectively. Defining $\Lambda := \Sigma_Z \Sigma_Z^{\mathsf{T}}/(n-1)$, the diagonal components $\lambda_{Z,i}(i = 1,...,d)$ of $\Lambda_Z$ are equal to the eigenvalues (PC variances) of PCA. That is, for the covariance matrix $S_Z = (Z - \overline{Z})(Z - \overline{Z})^{\mathsf{T}}/(n-1)$, $\lambda_{Z,i}(i = 1,...,d)$ satisfies the eigenvalue equation $S_Z u_{Z,i} = \lambda_{Z,i} u_{Z,i}$, where $u_{Z,i}$ is the $i$th column vector of $U_Z$. Furthermore, for the PCA transformation $Z^{\text{PCA}} = U_Z^{\mathsf{T}}(Z - \overline{Z})$, the $i$th eigenvalue corresponds to the variance of the $i$th principal component ($i$th PC variance), that is, $\text{Var}_i\left(z_{ij}^{\text{PCA}}\right) = \lambda_{Z,i}$. Because NVSN resolved COD3, the PCA transformation can divide the essential and noise parts by the first $\ell$ principal components and the others ($\ell$: threshold). Accordingly, we introduce the following modified eigenvalues

$$\widetilde{\lambda}_{Z,i} = \begin{cases} \lambda_{Z,i} - \dfrac{1}{d^{\text{PCA}} - i} \displaystyle\sum_{j=i+1}^{d^{\text{PCA}}} \lambda_{Z,j}, & i \le l \text{(essential part)}, \\[2mm] 0, & i > l \text{(noise part)}. \end{cases}$$

Here, $d^{\text{PCA}}$ is the dimension of PCA-projected data, that is, $d^{\text{PCA}} = \min\{n-1, d\}$. The transformation of the essential part (PC variance modification in Fig 3A) adopts Yata and Aoshima's method that modifies eigenvalues to converge to true eigenvalues (Yata & Aoshima, 2012). The transformation of the noise part (PC variance elimination in Fig 3A) eliminates the effect of the noise part by setting the eigenvalues to be zero. Here, we can optimize the threshold $\ell$ by using the fact that all the noise variances of the normalized data are one, as

$$\ell^{\text{opt}} = \min\left\{ k \in \{1,...,d\}; \sum_{i=k+1}^{d} \lambda_{Z,i} \le (d-k) \right\}.$$

Then, we define the denoised data $\tilde{Z} = (\tilde{z}_{ij})$ as

$$\widetilde{Z} = U_Z \widetilde{\Lambda}_{Z,\ell^{\text{opt}}}^{1/2} \Lambda_Z^{-1/2} U_Z^{\mathsf{T}}(Z - \overline{Z}) + \overline{Z}.$$

Here, $\widetilde{\Lambda}_{Z,\ell^{\text{opt}}}^{1/2}$ and $\Lambda_Z^{-1/2}$ are diagonal matrices with diagonal entries $\tilde{\lambda}_{Z,1}^{1/2},...,\tilde{\lambda}_{Z,d}^{1/2}$ ($\ell = \ell^{\text{opt}}$) and $\lambda_{Z,1}^{-1/2},...,\lambda_{Z,d}^{-1/2}$, respectively. We treat $(\Lambda_Z^{-1/2})_{ii} = 0$ when $\lambda_{Z,i} = 0$. The denoised data $\tilde{Z}$ satisfies $S_{\tilde{Z}} u_{Z,i} = \tilde{\lambda}_{Z,i} u_{Z,i}$ for $i = 1,...,d$ and $\overline{\tilde{Z}} = \overline{Z}$. That is, the eigenvalues are modified to $\tilde{\lambda}_{Z,i}$, whereas the principal component structure and the center values are preserved. Finally, we obtain the denoised scaled data $\tilde{x}_{ij}$ and UMI count data $\tilde{c}_{ij}$ by applying the inverse functions as

$$\tilde{x}_{ij} = f_{E_i(x_{ij}/t_j)}^{-1}(\tilde{z}_{ij}) := \sqrt{E_i(x_{ij}/t_j)}\,\tilde{z}_{ij},$$

$$\tilde{c}_{ij} = t_j \tilde{x}_{ij}.$$

In the computation, the noise variance $E_i(x_{ij}/t_j)$ is evaluated by

$$E_i(x_{ij}/t_j) \approx \alpha_i := \frac{1}{n} \sum_{j=1}^{n} \frac{x_{ij}^{\star}}{t_j^{\star}}.$$

Here, the star symbol $\star$ indicates the sample values. Then, we generate the normalized value as $z_{ij}^{\star} = f_{\alpha_i}(x_{ij}^{\star})$.

## Classification of genes

We define the following classification of genes:

$$\text{gene } i \text{ is } significant \overset{\text{def}}{\Leftrightarrow} x_{ij}^{\text{true}} \ne \text{constant for } j = 1,...,n,$$

$$\text{gene } i \text{ is } non\text{-}significant \overset{\text{def}}{\Leftrightarrow} x_{ij}^{\text{true}} = \text{positive constant for } j = 1,...,n,$$

$$\text{gene } i \text{ is } silent \overset{\text{def}}{\Leftrightarrow} x_{ij}^{\text{true}} = 0 \text{ for } j = 1,...,n.$$

From this definition, the significant genes capture cell-specific features, whereas the nonsignificant genes do not identify cell differences. The silent genes are those having no function. We denote the index sets of significant, nonsignificant, and silent genes as $I_{\text{sig}}$, $I_{\text{non-sig}}$, and $I_{\text{silent}}$, respectively. Then, from Equation (1), we obtain

$$\text{Var}(z_{ij}) \begin{cases} > 1, & i \in I_{\text{sig}}, \\ = 1, & i \in I_{\text{non-sig}}, \\ = 0, & i \in I_{\text{silent}}. \end{cases} \tag{2}$$

Thus, we can classify genes from the variances of normalized data by NVSN, which hereafter denotes NVSN variances. The classification allows us to determine whether a gene defines a difference among cell populations in scRNA-seq data. Moreover, this classification is also used for the applicability of RECODE discussed in the next section.

## Applicability of RECODE

In principle, RECODE can be used for any scRNA-seq data and reduces noises induced by random sampling. However, the output data may still retain noises caused by other technical problems such as amplification. To examine the effects of denoising, we can judge the applicability of RECODE by observing the NVSN variance distribution.

We can first show that the empirical NVSN variances of expressed genes are lower bounded by one

$$s_{Z^{\star},i}^2 = \frac{1}{n-1} \sum_{j=1}^{n} \left( z_{ij}^{\star} - \overline{z}_i^{\star} \right)^2 \ge 1, \quad i = 1,...,d, \tag{3}$$

under Equation (2), where $\bar{z}_i^\star$ is the empirical mean of the gene $i$. According to this property, we classify the scRNA-seq data as follows:

Class A (strongly applicable): The NVSN variances $s^2_{z^\star,i}$ satisfy condition (3) and

$$s^2_{z^\star,i} \approx 1 \text{ for most genes } i.$$

Class B (weakly applicable): The NVSN variances $s^2_{z^\star,i}$ satisfy condition (3) and

$$s^2_{z^\star,i} \gg 1 \text{ for most genes } i.$$

Class C (inapplicable): The NVSN variances $s^2_{z^\star,i}$ do not satisfy condition (3).

Here, we recall that most genes are not directly related to cell identifications. In our setting, this fact indicates that the number of $I_{\text{non-sig}}$ (nonsignificant genes) is much larger than that of $I_{\text{sig}}$ (significant genes), leading to the second requirement in class A. It should also be noted that noise variances generally increase when there exist other noise effects in addition to random samplings, such as amplification errors. The additional noises cause the NVSN variances $s^2_{z^\star,i}$ to take large values away from one, as addressed in class B. Based on this discussion, the data in class A match our model, and hence, we can expect that RECODE appropriately removes noise. On the other hand, the data in class B imply that other noises may be mixed in with the random sampling noise, and accordingly, RECODE may only work for the partial removal of noise. Finally, because the data in class C do not follow our model, we cannot expect noise removal there.

**Simulation data creation**

We created the simulation data in Figs 1, 3, S1, and S7 based on the Splatter algorithm (Zappia et al, 2017). We set 1,000 cells, 20,000 genes, and 200 differential expressed genes identifying the four groups. We used the trended cell mean $\lambda^{\text{SPL}}_{ij}$ as the reference data ($c^{\text{true}}_{ij} = \lambda^{\text{SPL}}_{ij}$), where the symbol SPL denotes the Splatter variables. To represent the low detection rate, we created the observed count as $c_{ij} = \text{Poisson}\left(r\lambda^{\text{SPL}}_{ij}\right)$, where $r$ denotes the detection rate set as 0.1. We set the other parameters as follows: $\alpha^{\text{SPL}} = 0.3$, $\beta^{\text{SPL}} = 0.6$, $\pi^{O,\text{SPL}} = 0$, $\mu^{O,\text{SPL}} = 4$, $\sigma^{O,\text{SPL}} = 0.5$, $\mu^{L,\text{SPL}} = 11$, $\sigma^{L,\text{SPL}} = 0.2$, $\phi^{\text{SPL}} = 0$, $df^{\text{SPL}}_0 = 60$, $x^{\text{SPL}}_0 = 0$, and $k^{\text{SPL}} = -1$.

**Collection of human iPSCs and PGCLCs**

All the experiments on the induction of hPGCLCs from hiPSCs were approved by the Institutional Review Board of Kyoto University and were performed according to the guidelines of the Ministry of Education, Culture, Sports, Science, and Technology (MEXT) of Japan.

The culture of human iPSCs and the induction of PGCLCs were performed as described previously (Sasaki et al, 2015; Kojima et al, 2017, 2021). The 585B1 BTAG (585B1-868, bearing 2A-tdTomato and 2A-EGFP at *BLIMP1* and *TFAP2C* loci) hiPSCs were maintained in StemFit AK03N medium (Ajinomoto) on cell culture plates coated with iMatrix-511 (Nippi). The medium was changed every other day. For the passaging or the induction of differentiation, the cells were treated with a one-to-one mixture of TrypLE Select (Life Technologies) and 0.5 mM EDTA/PBS to dissociate into single cells, and $10\mu M$ of a ROCK inhibitor (Y-27632; Wako Pure Chemical Industries) was added for 24 h after plating.

For the induction of hPGCLCs, hiPSCs were first plated at a density of $5 \times 10^4$ cells/cm$^2$ onto a fibronectin (FC010; Millipore)-coated plate. The cells were then cultured in GK15 medium (GMEM with 15% KSR, 0.1 mM NEAA, 2 mM L-glutamine, 1 mM sodium pyruvate, penicillin–streptomycin, and 0.1 mM 2-mercaptoethanol) supplemented with 50 ng/ml activin A (R&D Systems), 3 $\mu M$ CHIR99021 (TOCRIS), and $10\mu M$ of Y-27632 (Wako Pure Chemical Industries) for 44–48 h. Next, the cells were dissociated into single cells with TrypLE Select and aggregated in a low cell-binding V-bottom 96-well plate (Greiner) at 5,000 cells per well in 100 $\mu l$ of GK15 medium supplemented with 200 ng/ml BMP4 (R&D Systems), 100 ng/ml SCF (R&D Systems), 50 ng/ml EGF (R&D Systems), 1,000 U/ml LIF (Millipore), and 10 $\mu M$ of Y-27632 to be induced into hPGCLCs.

For the sample collection from iPSCs, the cells were suspended in the same manner as described above. For the aggregates of PGCLC induction, the aggregates were collected on the designated days of induction day 4, washed once in PBS, and dissociated with 0.25% trypsin–EDTA for 10–15 min at 37°C with gentle pipetting every 5 min. Trypsin was neutralized with a 5× volume of 10% FBS in DMEM. Before proceeding with the scRNA-seq data analysis, the live BTAG double-positive single cells were sorted using a FACSAria III system (BD Biosciences) by FSC-SSC and EGFP-tdTomato gating with DRAQ7 (ab109202) staining. To make an aliquot of the mixture of iPSCs and day 4 BTAG double-positive cells, equal numbers of cells were mixed before the scRNA-seq data analysis.

**10X scRNA-seq data acquisition**

ScRNA-seq libraries of 10X data were generated using the 10X Genomics Chromium Controller (10X Genomics) and Chromium Single Cell 3′ Reagent Kits v3.1 according to the manufacturer's instructions. Reverse transcription, cDNA amplification, and sample indexing were performed using an Eppendorf Mastercycler. The final libraries were quantified using a KAPA library quantification kit (KK4824), and the fragment size distribution of the libraries was determined using a LabChip GX DNA high sensitivity kit (Perkin Elmer). Pooled libraries were then sequenced using a NovaSeq 6000 Illumina Sequencer with a NovaSeq 6000 S1 Reagent Kit (100 Cycles, 20012865).

**Mapping of scRNA-seq data**

For human PBMC CITE-seq data, the filtered count matrix data were downloaded from the 10X Genomics demo data site (see Data Availability in the Materials and Methods section for details).

For the other 10X 3′ scRNA-seq data, CellBench data by Tian et al (2019), the hPGCLC induction data by Chen et al (2019), the hiPSC/hPGCLC mixture data generated in this study, and the mouse gastrula data by Pijuan-Sala et al (2019), raw files were processed with CellRanger 6.0.1 using default mapping arguments. Reads were mapped to the mouse genome (GRCm38.p6) or human genome (GRCh38.p12), respectively.

For the Human Cell Atlas project data by Mereu et al (2020) and Hagemann-Jansen et al (2020), the 10X data were processed with CellRanger 6.0.1 using default mapping arguments, and the Quartz-seq, Smart-seq2, and Smart-seq3 data were processed with zUMI v2.9.7 (Parekh et al, 2018) and STAR v2.7.3 (Dobin et al, 2013) according to the original papers (Hagemann-Jensen et al, 2020; Mereu et al, 2020). Briefly, reads were mapped onto a simple human genome (GRCh38.p12) and onto a custom genome consisting of human (GRCh38.p12), dog (CanFam3.1), and mouse (GRCm38.p6) genomes with chromosome and gene names headed by "hs_," "mm_," and "cf_," respectively, to identify the organisms.

### Mapping of bulk RNA-seq data

For the hPGCLC and hiPSC data generated by the 10X 3' scRNA-seq, conversion of the read data into expression levels was performed as described previously (Nakamura et al, 2015, 2016; Kojima et al, 2017). The reads were mapped on the human genome (GRCh38.p12). The 3' RNA-seq reads only the 3-prime ends of transcripts so that the expression levels were calculated as genes (Entrez genes) but not transcripts and reads per million-mapped reads (RPM) were calculated for the expression levels. Then the RPM expression data matrix was transformed to $\log_2(RPM+1)$ and used for the downstream data analysis.

For the five adenocarcinoma cell line data generated by conventional full-length RNA-seq (Counterparts to CellBench, [Holik et al, 2017]), the reads were mapped on the human genome (GRCh38.p12), and the expression levels were calculated as fragments per kilo-base million-mapped reads (FPKM). Then, the FPKM expression data matrix was transformed to $\log_2(FPKM+0.1)$ and used for the downstream data analysis.

### ScRNA-seq data quality check and preprocessing

In the quality check of scRNA-seq data, the cells with low nUMIs mapped on genes, low nGenes, high percentages of UMIs on mitochondrial genes, and the multiplets judged by Scrublet (Wolock et al, 2019) were filtered out. In addition, for 10X CITE-seq data of PBMCs, the cells with low nUMIs mapped on the protein index were filtered out. For the Human Cell Atlas data, first, the multiplets between different species were omitted according to the ratio of UMIs mapped on the human genome. Then, using the data mapped onto the human genome (GRCh38.p12), the cells with low nUMIs mapped on genes, low nGenes, and high percentage UMIs of mitochondrial genes were filtered out. See Figs S2 and S8 for detailed criteria. For preprocessing of the expression values, each UMI value was divided by the total UMI count of the cell, multiplied by the scale factor parameter (10k = 10,000) and then $\log_2$-transformed after adding a pseudo expression value of 1 to avoid the incorrect value ($\log_2 0$) [$\log_2(ss10k + 1)$].

### Applying RECODE

RECODE is applied to the quality-checked UMI count matrices using Python version 3.7.4 of Anaconda. The output matrices were log-normalized ($\log_2(ss10k + 1)$) for downstream data analyses.

### Applying imputation methods

The imputation methods were performed with default settings according to the respective program manuals. SAVER (Huang et al, 2018), scImpute (Li & Li, 2018), and ENHANCE (Wagner et al, 2019 Preprint) were applied to the UMI count matrices, and the output matrices were log-normalized ($\log_2(ss10k + 1)$). In contrast, MAGIC (van Dijk et al, 2018), DrImpute (Gong et al, 2018), and ALRA (Linderman et al, 2022) were applied to the log-normalized UMI count matrices ($\log_2(ss10k + 1)$). Minus values in their output matrix were forcibly set as 0.

### Analysis of scRNA-seq data

The following analysis was performed using Microsoft-R software version 4.0.2 with the parallelDist, amap, Rtsne, gplots, uwot, cluster, qvalue, and Seurat packages. UHC was performed using the hclust and parDist functions with Euclidean distances and Ward's method (ward.D2). t-SNE was performed using the Rtsne function with perplexity = 30 and genes with expression ($\log_2(ss10k + 1) > 0$) in at least one cell. UMAP was performed using the umap function with n_neighbors = 10, min_dist = 0.5, and genes with expression ($\log_2(ss10k + 1) > 0$) in at least one cell. The cell types of human PBMC CITE-seq, CellBench, and the hiPSC/hPGCLC mixture datasets were defined by using external data as described below (Figs S3–S5). Some of the ENHANCE-treated cells became identical to each other, and thus, those cells were removed ahead of clustering to avoid computation errors. Note that, for analyses other than Seurat, we performed these multi-variance analyses without any other dimension reduction steps, such as selecting HVGs or aggregation of gene information by PCA. The Silhouette scores were calculated using Silhouette function with Euclidean distances calculated by parDist function. To identify DEGs, the wilcox.test and kruskal.test functions followed by the q value function were used to calculate the P-value and false discovery ratio, respectively.

For analyses with Seurat, importation of the raw UMI count matrix, $\log_2$ normalization [$\log_2(ss10k + 1)$], scaling, selection of HVGs (2,000 genes), and PCA were performed by NormalizeData, FindVariableFeatures, ScaleData, and RunPCA functions with default settings according to the tutorial on the Seurat website (https://satijalab.org/seurat/articles/pbmc3k_tutorial.html). Selecting the significant PCs was done based on the results of the jackstraw function. The following analysis (UHC, UMAP, and t-SNE) were performed with the parallelDist, Rtsne, and uwot packages using the PC scores of the significant PCs suggested by jackstraw (mouse gastrula data E6.5; PC26, E7.5; PC31).

### Analysis of Drop-seq and RNA-FISH data

The analysis of Drop-seq and RNA-FISH data (Fig 7) was performed as follows. The cells with nUMI less than 500 or greater than 20,000 were removed following the previous study (Huang et al, 2018), but all the expressed genes (28,537 genes) were used. For both the data without preprocessing and that with preprocessing by RECODE, SAVER, and MAGIC, we aligned the averages of each gene with RNA-FISH data to fairly compare the distributions in Fig 7A. The $k$ th order moments of gene $i$ in Fig 7B are defined as $M_k := n^{-1}\sum_{j=1}^{n}(x_{ij} - \overline{x}_i)^k$,

where $x_{ij}$ and $\bar{x}_j$ are the log-scaled gene expression value of gene $i$ on cell $j$ and the average value of gene $i$, respectively. The relative error of the $k$ th order moment is defined as $\left|M_k^{\text{FISH}} - M_k^{\text{Drop}}\right|/\left|M_k^{\text{FISH}}\right|$, where $M_k^{\text{FISH}}$ and $M_k^{\text{Drop}}$ are the $k$ th order moments computed by RNA-FISH and Drop-seq data, respectively.

### Identification of cell types in human PBMC CITE-seq, CellBench, and the hiPSC/hPGCLC mixture dataset

For cell annotations in human PBMC CITE-seq data, raw UMI counts of protein expression (CD3, CD4, CD8A, CD14, CD15, CD16, CD19, CD25, CD56 CD137, CD45RA, and CD45RO) were processed by dividing each UMI by the total UMI count of the proteins in the cell (nUMI_protein), multiplying it by the median of nUMI_protein, and then $\log_2$-transforming it after adding a pseudo expression value of 1 to avoid the incorrect value ($\log_2 0$) [$\log_2(\text{ss.med} + 1)$]. The cell clustering and annotations were determined according to the UHC analysis and the expression pattern of the protein expressions. The UHC and UMAP were performed with the parameters described above.

For the annotation of the CellBench dataset, first, the signature genes of each cell type were calculated using the corresponding bulk RNA-seq data (Holik et al, 2017) (174 genes, $\log_2$F.C. [mean of target cells versus mean of the others] >3 and mean of $\log_2$(FPKM+0.1) of the target cells >4). The cell clustering of Cell-Bench 10X data was performed according to the UHC analysis using the $\log_2$(ss10k + 1) values of the 174 signature genes. Finally, the correlation between the means of the clusters of 10X CellBench data and bulk RNA-seq data were computed and the cells of CellBench were annotated according to the results of the correlation analysis.

For the annotation of the hiPSC/hPGCLC mixture data, first, the signature genes of each cell type were calculated using the corresponding bulk RNA-seq data (Kojima et al, 2017) (442 genes, $\log_2$F.C. [mean of hiPSC versus mean of hPGCLC] >3 and mean of $log_2$(RPM + 1) > 4). The clustering of 10X scRNA-seq data was performed according to the UHC analysis using the $\log_2$(ss10k + 1) values of the 442 signature genes. We annotated the clusters with the expressions of *SOX17*, *TFAP2C*, *PRDM1*, and *NANOS3* and with *SOX2* and *DNMT3B* as hPGCLC and hiPSC, respectively. The clusters with both PGCLC and iPSC marker expressions should be doublets, so they were removed ahead of the following analysis. Note that we assumed that the effect of COD is negligible for such a small number of genes (dimensions).

### In silico dilution analysis

For the assay in Fig 8, 100, 20, 10, 5, 3, 1 hiPSCs or H2228 cells in the hiPSC/hPGCLC mixture or the CellBench data were picked randomly and mixed with the other remaining cells (Fig 8A and B). Then the expression matrices were applied for RECODE and other imputation methods in the same manner as described above.

### Evaluation of run-time, memory, and scalability

We used scRNA-seq data with 36,694 cells ("uclc2" series of GSE140021, Chen et al, 2019, see Table S1 for details) to evaluate the run-time, memory usage, and scalability of RECODE, SAVER, and

MAGIC (Fig 9). We created datasets by randomly sampling 200, 500, 1,000, 2,000, 5,000, 10,000, 20,000, and 30,000 cells from the original data. The run-times were measured by the time.time function on python for RECODE and the tictoc package on R for SAVER and MAGIC. The memory usages were measured by the maximum memory usage during the job by the Grid Engine system (UNIVA Grid Engine) of our computation server. The scalabilities were calculated by the linear regression of run-times for 10,000–30,000 cells. The brief specs of the computation node used for this assay are CPU: dual Xeon Gold 6146 (12 cores × 2, 3.2 GHz, without hyperthreading technology) and memory: DDR4-2666, 512 GB.

## Data Availability

The accession numbers used in this study are as follows. For the scRNA-seq data, the hiPSC/hPGCLC mixture data generated in this study (GSE175525); human PBMC CITE-seq data (demo data from 10X genomics: *10k PBMCs from a Healthy Donor—Gene Expression with a Panel of TotalSeq-B Antibodies*) (https://www.10xgenomics.com/resources/datasets/10-k-pbm-cs-from-a-healthy-donor-gene-expression-and-cell-surface-protein-3-standard-3-0-0), the Cell-Bench data (GSE118767) (Tian et al, 2019), the hPGCLC induction (GSE140021) (Chen et al, 2019), the Drop-seq data (GSE99330) (Torre et al, 2018), datasets in the Human Cell Atlas project (10X 3′ scRNA-seq, Drop-seq, Quartz-seq, Smart-seq2, Smart-seq3) (GSE133549, E-MTAB-8735) (Hagemann-Jensen et al, 2020; Mereu et al, 2020), and the mouse gastrula data (E-MTAB-6967) (Pijuan-Sala et al, 2019).

For the external data, the bulk RNA-seq data of five adenocarcinoma cell lines for CellBench (GSE86337) (Holik et al, 2017), the bulk RNA-seq data of iPSCs and the purified hPGCLCs (GSE99350) (Kojima et al, 2017), and the smFISH data count data were obtained from https://www.dropbox.com/sh/g9c84n2torx7nuk/AABZei_VVpcfTUNL7buAp8z-a?dl=0 (Torre et al, 2018). See Table S1 for details.

### Code availability

The python and R codes of RECODE are available at https://github.com/yusuke-imoto-lab/RECODE.

## Supplementary Information

## Acknowledgements

We are grateful to Dr. Kazuyoshi Yata for the valuable discussions and comments and to Y Nagai, N Konishi, E Tsutsumi, and M Kawasaki of the Saitou Laboratory for their technical assistance. We thank the Single-Cell Genome Information Analysis Core (SignAC) in ASHBi for the RNA sequence analysis. This work was sponsored by the World Premier International Research Center Initiative (WPI). This work was also sponsored by a JST PREST Grant (no. JPMJPR2021 to Y Imoto), a JSPS Grant-in-Aid for Early-Career Scientists (no. 18K14714 to T Nakamura), a MEXT Grant-in-Aid for Transformative Research Areas B (no. 20H05761 to T Nakamura), a JST CREST

Mathematics Grant (no. 15656429 to Y Hiraoka), a JST MIRAI Program Grant (no. 22682401 to Y Hiraoka), an AMED-CREST Grant (no. JP19gm1310002h to Y Hiraoka), and a Grant-in-Aid for Specially Promoted Research from JSPS (nos. 17H06098 and 22H04920 to M Saitou).

## Author Contributions

Y Imoto: conceptualization, data curation, software, formal analysis, funding acquisition, validation, investigation, visualization, methodology, writing—original draft, review, and editing, and code implementation.

T Nakamura: conceptualization, data curation, formal analysis, funding acquisition, validation, investigation, visualization, methodology, and writing—original draft, review, and editing.

EG Escolar: investigation and writing—original draft.

M Yoshiwaki: investigation and writing—original draft.

Y Kojima: data curation and investigation.

Y Yabuta: investigation.

Y Katou: investigation and code implementation.

T Yamamoto: conceptualization.

Y Hiraoka: conceptualization, supervision, funding acquisition, investigation, project administration, and writing—original draft, review, and editing.

M Saitou: conceptualization, supervision, funding acquisition, project administration, and writing—original draft, review, and editing.

## Conflict of Interest Statement

Y Imoto, T Nakamura, Y Hiraoka, and M Saitou are inventors on patent applications relating to RECODE filed by Kyoto University.

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
