## [Reviewer comments · Life Science Alliance]

Life Science Alliance

Resolution of the curse of dimensionality in single-cell RNA sequencing data analysis

Mitinori Saitou, Yusuke Imoto, Tomonori Nakamura, Emerson G. Escolar Escolar, Michio Yoshiwaki, Yoji Kojima, Yukihiro Yabuta, Yoshitaka Katou, Takuya Yamamoto, and Yasuaki Hiraoka

DOI: <https://doi.org/10.26508/lsa.202201591>

Corresponding author(s): Mitinori Saitou, Institute for the Advanced Study of Human Biology, Kyoto University and Yasuaki Hiraoka, Institute for the Advanced Study of Human Biology, Kyoto University

Review Timeline:

Submission Date:	2022-07-05
Editorial Decision:	2022-07-07
Revision Received:	2022-07-08
Editorial Decision:	2022-07-13
Revision Received:	2022-07-15
Accepted:	2022-07-18

Scientific Editor: Novella Guidi

Transaction Report:

Please note that the manuscript was previously reviewed at another journal and the reports were taken into account in the decision-making process at Life Science Alliance.

Reviewer #1 Review

Report for Author:

In my opinion, the authors have improved the manuscript, but it is still lacking a suitable orthogonal experimental validation. Here are my comments:

1) The authors argue that RNA-FISH does not represent a perfect ground truth and that RNA-FISH still has a high dropout rate of 20%. However, scRNA-seq methods, particularly the UMI- and microfluidics-based methods on which the authors focus have dropout rates of 90-95%. So surely RNA-FISH is giving a more accurate picture of the true transcript abundance distribution than scRNA-seq. Indeed, it is frequently used for in situ validation of findings from scRNA-seq. In any case, the authors now show a panel with some smFISH analysis (Figure EV4I). However, as far as I can tell, this analysis is not described or discussed in the text. Furthermore, it only compares unprocessed scRNA-seq and RECODE-processed scRNA-seq to smFISH, but does not provide any information on how well RECODE performs in this comparison vs. alternative imputation methods.

2) While the authors dispute the utility of RNA-FISH as a comparator for scRNA-seq, they use CITE-seq/FACS-based comparisons more extensively. Of course, these methods are measuring surface protein abundance (not even total protein abundance) rather than mRNA abundance, and so I would argue that this is an even more problematic comparison than RNA-FISH. While I think this analysis is still somewhat valuable, the presentation should be more quantitative. For example, the authors point out that the RECODE results in the biaxial plots in Fig. 6E look more similar to the protein-level analysis than alternative methods. However, none of the scRNA-seq results really look highly similar to the protein-level results, and so some quantification of similarity should be done. For example, do the RECODE data provide similar cell frequencies for the ~3 major cell populations shown in the protein-level analysis in comparison to alternative methods? Or is there some way of quantifying the 2D distributions shown in the biaxial plots?

Reviewer #2 Review

Report for Author:

This is a resubmission of the earlier version. The authors related the question of scRNA-seq data preprocessing to the curse of dimensionality and utilized the corresponding theories to develop a computational method RECODE. RECODE performs variance stabilization and shrinkage based on estimated noise level and includes applicability tests. The authors demonstrated its utility in a practical setting in addressing the curses and in cell clustering. RECODE was compared against several other methods in questions like cell clustering and differential gene expression.

This version is greatly improved and heavily revised from the last submission, especially in relating to the scRNA-seq preprocessing question, the comprehensiveness of evaluations, and better structuring. The applicability test focuses on the model assumption and is an advantage to most preprocessing methods. Still, the evaluations were primarily restricted to cell clustering and not many other preprocessing methods. Although a few major conclusions appear slightly overstated, they would become convincing after properly revising the claims. This manuscript provides helpful insights in decomposing variations in scRNA-seq data and in improving preprocessing. This could be beneficial for potential users of RECODE and statistical method developers for scRNA-seq.

I still have several concerns for this manuscript.

Major points

1. Quite a few statements in this paper are not sufficiently objective or even correct. They may appear evident or convincing to the authors but not conclusive, obvious, or sufficiently evidenced to the reader. A large proportion are newly added statements not in the last version. Several examples:

"RECODE does not involve dimension reduction..." PCA projection is an essential step in Fig 3A.

"RECODE shortens the legs of the dendrogram in UHC compared to those in the observed data and, accordingly, obtains more correct clustering." The consequential relation is not justified here. Correct clustering depends on groundtruth and cannot be determined with leg length alone.

"...RECODE extinguishes those noises and recovers the true expression values..." The evidence is insufficient for fully extinguishing those noises or for completely recovering true expression values. In Response 4, the authors provided evidences in several figures. However, they can only reflect reduction in total variance, not the variance contribution specifically from one noise type. These figures do not provide evidence for complete removal of the noise either.

The authors should fully review the manuscript to ensure every claim is appropriate to the level of evidence provided, and consider inviting independent feedbacks from friends.

2. Related to point 1. The authors should specify what "a general strategy for preprocessing" indicates, multiple purposes or applicability to different datasets? For the earlier, utility in more than one type of application is needed. Despite the extensive evaluation in clustering, differential expression needs more comprehensive evaluations such as in logFC and p-values, and with more methods.

3. Related to Response 6. Silhouette score has been used in Fig 5C. Can the authors provide Silhouette score or other quantitative metrics for Fig 5E and 6A?

4. Related to point 1. Although Fig 5FG are helpful in demonstrating RECODE's performance, they are insufficient to show the exact recovery of true expression values. True expression would require comparability between cells and between genes, and accurate absolute levels. Since only comparability between cells is demonstrated here, the author should consider rephrasing the claim. Also, what is relative expression?

5. The fairness of the comparison in Fig 6A remains unclear to me. Other methods do not include a prior dimension reduction step like PCA, but RECODE acts partly like a PCA dimension reduction. However all methods should have their expected up/downstream steps for best performance. How do other methods perform if a standard PCA step is placed between preprocessing and UMAP?

Minor points

6. The methods of RECODE have symbol errors and mixups, such as in the definition of t_j . The authors should double check all the symbols and equations in methods.

7. The classification of genes is actually derived from the assumption $\text{Var}_i(x_{ij}^{\text{true}}) \ll \text{Var}_i(x_{ij})$. However only x_{ij}^{true} is constant is mentioned, and the relation between gene classification and this assumption is not clear in the paper. Clarifying these two in methods would make RECODE's applicability more understandable.

8. Related to point 1. "...the current scRNA-seq data analysis may not capture the true gene-to-gene relations..." I suppose the authors meant scRNA-seq data analysis without preprocessing?

Reviewer #3 Review

Report for Author:

The authors have carefully considered feedback received and done major improvements in the clarity of explaining their method and in inclusion of additional experiments and benchmarks to support the validity of conclusions drawn.

The revised manuscript clearly demonstrates caveats in current best practices. The evaluation with simulated and real data is supporting key conclusions drawn regarding benefits of the proposed RECODE method against existing solutions. The method and its implementation is shared with the community. This work will contribute significantly conceptually and in the methodology available to analyze the single cell genomics that is now widely adopted in the field of biomedicine.

Minor comments:

The current Discussion is focused on additional elaboration of presented results and future directions but limited in placing the work in context of previous related work, or bringing up possible unexpected findings.

-The distinctions as made in figure panels on the types of approaches proposed in the field could be re-stated to summarize the context also in this section

-The suitability to 10X data is supported by data presented. Was it surprising that Smart-seq protocols gave different results?

-Batch effects: this is left for future work - however, would their method remedy some aspects of the challenge e.g would processing each sample within a batch with RECODE, followed by inclusion of existing batch normalization methods or inclusion of batch in DEG statistical models downstream represent a viable practical solution?

July 7, 2022

Re: Life Science Alliance manuscript #LSA-2022-01591-T

Mitinori Saitou
Institute for the Advanced Study of Human Biology, Kyoto University
Yoshida-Konoe-cho, Sakyo-ku
Kyoto 606-8501
Japan

Dear Dr. Saitou,

Thank you for submitting your manuscript entitled "Resolution of the curse of dimensionality in single-cell RNA sequencing data analysis" to Life Science Alliance. The manuscript was assessed by expert reviewers at another journal, and then revised by the authors accordingly. The revised manuscript has been then seen by the original reviewers and although overall positive, some important issues remained. The reviewer comments were assessed at LSA, and LSA editors deemed that the manuscript could be published at LSA provided the authors revise the manuscript and address all the final remaining comments.

Thank you for this interesting contribution to Life Science Alliance. We are looking forward to receiving your revised manuscript.

Sincerely,

B. MANUSCRIPT ORGANIZATION AND FORMATTING:

Rebuttal: LSA-2022-01591-T

We would like to sincerely thank the Reviewers for their evaluation. We have closely followed the Reviewers' comments in our revision. We hope you find the revised manuscript suitable for publication in *Life Science Alliance*.

Reviewers' comments:**Reviewer #1:**

In my opinion, the authors have improved the manuscript, but it is still lacking a suitable orthogonal experimental validation.

Response 1. We thank the Reviewer for acknowledging the improvements made in our revised manuscript. We would like the Reviewer to note that we have provided comprehensive validations of the performance of RECODE using diverse scRNA-seq datasets complemented with other measurements, such as RNA-FISH (Drop-seq data) [Fig EV4D, I (Fig 7 in the revised manuscript)] (Torre et al., 2018); protein expression (human PBMC CITE-seq and hPGCLC induction data) [Figs 5, 6, EV4B, E–H (Figs 5, 6, S9B, E–H in the revised manuscript)] (10X genomics demo data: 10k PBMCs from a Healthy Donor - Gene Expression with a Panel of TotalSeq™-B Antibodies) (Chen et al., 2019); bulk RNA-seq (CellBench and hiPSC/hPGCLC mixture data) [Figs 7, EV4A–C (Figs 8, S9A–C in the revised manuscript)] (Tian et al., 2019) (data generated for this study, GSE175525); and numerous *in situ* hybridization data [Fig 9 (Fig 10 in the revised manuscript)] (mouse gastrula scRNA-seq data) (Pijuan-Sala et al., 2019). Please also see **Response 2** below and Fig 7 in the revised manuscript for a quantitative, orthogonal validation of the performance of RECODE.

Here are my comments:

1) *The authors argue that RNA-FISH does not represent a perfect ground truth and that RNA-FISH still has a high dropout rate of 20%. However, scRNA-seq methods, particularly the UMI- and microfluidics-based methods on which the authors focus have dropout rates of 90-95%. So surely RNA-FISH is giving a more accurate picture of the true transcript abundance distribution than scRNA-seq. Indeed, it is frequently used for in situ validation of findings from scRNA-seq. In any case, the authors now show a panel with some smFISH analysis (Figure EV4I). However, as far as I can tell, this analysis is not described or discussed in the text. Furthermore, it only compares unprocessed scRNA-seq and RECODE-processed scRNA-seq to smFISH, but does not provide any information on how well RECODE performs in this comparison vs. alternative imputation methods.*

2) *While the authors dispute the utility of RNA-FISH is a comparator for scRNA-seq, they use CITE-seq/FACS-based comparisons more extensively. Of course, these methods are measuring surface protein abundance (not even total protein abundance) rather than mRNA abundance, and so I would argue that this is an even more problematic comparison than RNA-FISH. While I think this analysis is still somewhat valuable, the presentation should be more quantitative. For example, the authors point out that the RECODE results in the biaxial plots in Fig. 6E look more similar to the protein-level analysis than alternative methods. However, none of the scRNA-seq results really look highly similar to*

the protein-level results, and so some quantification of similarity should be done. For example, do the RECODE data provide similar cell frequencies for the ~3 major cell populations shown in the protein-level analysis in comparison to alternative methods? Or is there some way of quantifying the 2D distributions shown in the biaxial plots?

Response 2. Since these two comments are related, we have combined our responses. First, we would like to apologize to the Reviewer for the confusion that our prior response might have caused. We would like the Reviewer to note that we did not “dispute” the utility of RNA-FISH data; we simply pointed out the difficulty in defining the “true expression values” in single-cell gene expression even with RNA-FISH [e.g., see (Cao et al., 2021)]. It is known that RNA-FISH involves a significant degree of transcript dropouts, and the detection rate by RNA-FISH depends on many variables, such as the probe sequence, hybridization efficiency, higher-order structure of target RNAs, and detection limit by microscopy (Shah *et al.*, 2016) (**Response 3** to Reviewer #1 in the last round of communication). Indeed, the median expression level of *Gapdh* in the RNA-FISH data recommended by the Reviewer was ~360 copies/cell (Torre *et al.*, 2018), which is around one-tenth of the known *Gapdh* level in typical single cells (Nakamura et al., 2015). Moreover, while the RNA-FISH data examined the expression of 26 probes (genes) in total, the combinations of the probes used were somewhat variable in different sets of experiments (Torre *et al.*, 2018), and the data were not suitable for comprehensive comparisons with the scRNA-seq data, such as comparisons involving clustering and UMAP. Also, since the RNA-FISH data and the scRNA-seq data were generated from different single cells (Torre *et al.*, 2018), strict comparisons were not possible.

We therefore used the human PBMC CITE-seq data (10X genomics demo data: 10k PBMCs from a Healthy Donor - Gene Expression with a Panel of TotalSeq™-B Antibodies), which has a complexity level typical of *in vivo* human cell diversity, and the corresponding RNA- and protein-expression data obtained from the same single cells, to perform wider kinds of verifications (Figs 2, 5, and 6); indeed, such verifications were made at the request of the Reviewers in the initial set of reviews (**General Response 3** in the last round of communication).

Second, as kindly acknowledged by the Reviewer, we indeed applied RECODE to the RNA-FISH data as part of the verification of the RECODE performance, demonstrating that the RECODE-modified data distribution was much better than the non-preprocessed data distribution and was highly similar to the RNA-FISH data distribution. (Please see Fig EV4I (Fig S9I in the revised manuscript) in the previous manuscript: we sincerely apologize for having provided a statement on this result only in the legend to this figure panel. We expanded the description on the data in Fig S9 in the revised manuscript.) In addition, in response to the Reviewer’s comment, we have provided a further comparison of the performance of RECODE, SAVER, and MAGIC on the same RNA-FISH data. As a result, we found that, unlike in the RECODE-preprocessed data, both the SAVER- and MAGIC-preprocessed data show somewhat over-aggregated

distributions (Fig 7A in the revised manuscript). To gain quantitative insight into this aspect, we computed relative errors from the 2nd- to 6th-order moments of their distributions (the 2nd-order moment is the variance), which revealed lower error rates by RECODE than by SAVER and MAGIC in a majority of moments for these genes (Fig 7B in the revised manuscript). These findings unequivocally demonstrate that the quantitative performance of RECODE is better than those of SAVER and MAGIC.

Regarding the human PBMC CITE-seq data, in response to the Reviewer's comment, we have examined the frequencies of the $CD3D^+/CD19^-$, $CD3D^-/CD19^+$, and $CD3D^-/CD19^-$ populations in the scRNA-seq data without preprocessing and with preprocessing with RECODE, SAVER, and MAGIC, and have compared them with those of the CD3/CD19 protein expressions [Fig 6E (Fig 6F in the revised manuscript)]. Without preprocessing, the ratios of $CD3D^+/CD19^-$ and $CD3D^-/CD19^+$ were lower—while that of $CD3D^-/CD19^-$ cells was higher—than those in the protein plot. In contrast, preprocessing by RECODE, SAVER, and MAGIC all recovered the abundance ratios of three major populations, indicating that the suggested quantification does not distinguish the performance of the three methods. On the other hand, as described in the manuscript, the distribution of the $CD3D/CD19$ expression levels by MAGIC appeared somewhat aggregated and that by SAVER was sparse with low expression levels (Fig 6F in the revised manuscript), and the same trend has been more clearly demonstrated for the detection of rare cell populations shown in Fig 7 (Fig 8 in the revised manuscript). We have also shown the superiority of RECODE over SAVER and MAGIC in terms of the computational practicalities [Fig 8 (Fig 9 in the revised manuscript)].

We believe that, when taken together, the combined evidence (Figs 6F, 7, 8, and 9 in the revised manuscript) clearly demonstrates the superiority of RECODE over SAVER and MAGIC as a preprocessing method for scRNA-seq data. We provided relevant data and discussion in the revised manuscript (2nd paragraph of the “**Resolution of CODs in scRNA-seq data analysis by RECODE**” section, Fig 6F, 7 and the 3rd and 4th paragraphs of the “**Comparison of RECODE with imputation methods**” section).

Reviewer #2:

This is a resubmission of the earlier version. The authors related the question of scRNA-seq data preprocessing to the curse of dimensionality and utilized the corresponding theories to develop a computational method RECODE. RECODE performs variance stabilization and shrinkage based on estimated noise level and includes applicability tests. The authors demonstrated its utility in a practical setting in addressing the curses and in cell clustering. RECODE was compared against several other methods in questions like cell clustering and differential gene expression.

This version is greatly improved and heavily revised from the last submission, especially in relating to the scRNA-seq preprocessing question, the comprehensiveness of

evaluations, and better structuring. The applicability test focuses on the model assumption and is an advantage to most preprocessing methods. Still, the evaluations were primarily restricted to cell clustering and not many other preprocessing methods. Although a few major conclusions appear slightly overstated, they would become convincing after properly revising the claims. This manuscript provides helpful insights in decomposing variations in scRNA-seq data and in improving preprocessing. This could be beneficial for potential users of RECODE and statistical method developers for scRNA-seq.

Response 1. We would like to sincerely thank the Reviewer for his/her overall positive comments on our manuscript.

The Reviewer commented that “*Still, the evaluations were primarily restricted to cell clustering and not many other preprocessing methods.*” However, we would like the Reviewer to note that, in addition to cell clustering, the previous manuscript included various other evaluations, such as analyses of the PC contribution rate [Figs 5B and EV4B (S9B in the revised manuscript)], Silhouette score [Figs 5C and EV4C (S9C in the revised manuscript)], independence of total UMI counts [Figs 5D and EV4D (S9D in the revised manuscript)], gene-expression profiles [Figs 5F–H, 6C–E, 9A, EV4E–H (Figs 5F–H, 6D–F, 10A, S9E–H in the revised manuscript), and 7A in the revised manuscript], variance distribution [Fig 6B (Fig 6C in the revised manuscript)], detection of rare cell populations [Fig 7 (8 in the revised manuscript)], computational practicality [Fig 8 (9 in the revised manuscript)], and DEG [Figs 9C and E (Figs 10C and E in the revised manuscript)]. Please also see **Response 2** to Reviewer #1 and Fig 7 in the revised manuscript for a quantitative, orthogonal validation of the superiority of RECODE.

Please also note that the primary aim of our manuscript is to develop a methodology to resolve the curse of dimensionality (COD) associated with the analysis of noisy high dimensional data based on mathematical theories and not to compare RECODE with all other preprocessing methods. Based on the evaluation of various imputation methods from previous publications (Andrews and Hemberg, 2018; Hou et al., 2020), we have selected six well-rated imputation methods from different categories (Lähnemann et al., 2020) and one widely-used scRNA-seq data analysis platform based on dimension reduction (Seurat) for comparison, and demonstrated the superior performance of RECODE. Finally, please note that we have also compared RECODE with Randomly as suggested by Reviewer #1 (**Response 2** to Reviewer #1 and Figure for Reviewers, #1 in the last round of communication). We therefore believe that we have performed an appropriate and comprehensive comparison with other preprocessing methods.

I still have several concerns for this manuscript.

Major points

1. Quite a few statements in this paper are not sufficiently objective or even correct. They

may appear evident or convincing to the authors but not conclusive, obvious, or sufficiently evidenced to the reader. A large proportion are newly added statements not in the last version. Several examples:

"...RECODE extinguishes those noises and recovers the true expression values..." The evidence is insufficient for fully extinguishing those noises or for completely recovering true expression values. In Response 4, the authors provided evidences in several figures. However, they can only reflect reduction in total variance, not the variance contribution specifically from one noise type. These figures do not provide evidence for complete removal of the noise either.

Response 2. Considering the difficulty/impossibility in knowing “true expression values” in single-cell gene expression [e.g., see (Cao *et al.*, 2021)], we agree with the Reviewer that this is an overstatement. We have revised the statement as follows:

“...RECODE significantly reduces those noises and recovers the gene-expression values, even for lowly expressed genes...” (1st paragraph of the **Discussion** section).

Likewise, defining “true noise values” is also impossible. We have therefore used the reduction of expression-level variances of housekeeping genes, which are expected to consist mainly of experimental noise, as an indicator of noise reduction. We hope this approach meets the approval of the Reviewer.

"RECODE does not involve dimension reduction..." PCA projection is an essential step in Fig 3A.

Response 3. This comment is based on a misunderstanding of RECODE by the Reviewer. Following the noise-variance-stabilizing normalization (NVSN), RECODE uses PCA to divide the normalized data into essential and noise parts for PC variance modification and elimination, respectively, and subsequently combines these two parts to preserve all gene information (all dimensions) for data output (Fig 3A and **Materials and Methods**). Thus, clearly, RECODE is not categorized as a dimension reduction method.

RECODE shortens the legs of the dendrogram in UHC compared to those in the observed data and, accordingly, obtains more correct clustering." The consequential relation is not justified here. Correct clustering depends on groundtruth and cannot be determined with leg length alone.

Response 4. We would like to thank the Reviewer for pointing out that this statement was somewhat misleading. Indeed, we did not intend to suggest that the length of the “legs” of clustering reflects the accuracy of clustering. We meant to state that RECODE not only shortens the legs of the UHC dendrograms, but also successfully recovers the true cell clustering, as shown in the upper color bar under the dendrogram in Fig 3B. We have revised the statements as follows in the revised manuscript:

“RECODE shortens the legs of the dendrogram in UHC compared to those in the observed data and at the same time, achieves more correct clustering (Fig 3B)” [The 2nd paragraph of the section “**Resolution of the curse of dimensionality (RECODE)**”].

The authors should fully review the manuscript to ensure every claim is appropriate to the level of evidence provided, and consider inviting independent feedbacks from friends.

Response 5. In response to the Reviewer’s comment, we conducted a comprehensive review of all claims in the manuscript, including the solicitation of input from colleagues.

2. Related to point 1. The authors should specify what "a general strategy for preprocessing" indicates, multiple purposes or applicability to different datasets? For the earlier, utility in more than one type of application is needed. Despite the extensive evaluation in clustering, differential expression needs more comprehensive evaluations such as in logFC and p-values, and with more methods.

Response 6. We thank the Reviewer for this comment. We would like to point out that we have provided relevant data in the **Supplementary text**, along with the following statement in the last paragraph of the **Discussion**: “Importantly, RECODE does not impose any assumptions on the data types for their application; that is, RECODE is applicable to other sequencing data created by a similar platform. For example, with an appropriate preprocessing, the spatial gene expression and scATAC-seq data with 10X Chromium are also categorized as Class A (strongly applicable) (Sections 5.3 and 5.4 in the Supplementary text)”.

We have shown various drawbacks of the imputation methods, including worse clustering performance (ENHANCE, scImpute, DrImpute; Fig 6A), worse Silhouette score (scImpute, DrImpute; Fig 6B in the revised manuscript), failure to mitigate the variance of housekeeping genes [scImpute, DrImpute, ALRA; Fig 6B (6C in the revised manuscript)], inferior recovery of the gene-to-gene relationships (SAVER, MAGIC; Fig 7 in the revised manuscript) (please see **Response 2** to Reviewer #1 for more details), loss of the properties of rare cell populations [SAVER, MAGIC; Fig 7 (8 in the revised manuscript)], and higher computational cost/inferior scalability [SAVER, MAGIC; Fig 8 (9 in the revised manuscript)]. We therefore decided to evaluate the performance of RECODE for the detection of differentially expressed genes (DEGs) as compared to that of preprocessing in Seurat, one of the most widely used scRNA-seq data analysis platforms based on dimension reduction (Hao *et al.*, 2021; Satija *et al.*, 2015). As a result, we found that many previously reported DEGs were successfully captured in RECODE-preprocessed but not Seurat-preprocessed data. These results are shown in Fig 9 (10 in the revised manuscript)

Taking all these data into account and the Reviewer’s suggestion that the term “general” may be too strong, we changed the word “general” to “powerful” in the revised

manuscript (**Abstract**, the 4th paragraph of the **Introduction**, the 6th paragraph of the **Discussion**).

3. *Related to Response 6. Silhouette score has been used in Fig 5C. Can the authors provide Silhouette score or other quantitative metrics for Fig 5E and 6A?*

Response 7. In response to the Reviewer's comment, we have provided the Silhouette scores for human PBMC CITE-seq data without preprocessing and with preprocessing with RECODE and the other imputation methods in Fig 6B of the revised manuscript. The scores with RECODE, SAVER, MAGIC, ENHANCE, and ALRA were improved over those with no preprocessing, and the scores with MAGIC and ENHANCE were even higher than those by the other methods, which might have been due to the aggregation of expression values with these methods, as shown in Figs 6A (for ENHANCE), 6F, 7, and 8 (for MAGIC) in the revised manuscript. We have provided relevant statements in the revised manuscript (2nd paragraph of the "**Comparison of RECODE with imputation methods**" section).

4. *Related to point 1. Although Fig 5FG are helpful in demonstrating RECODE's performance, they are insufficient to show the exact recovery of true expression values. True expression would require comparability between cells and between genes, and accurate absolute levels. Since only comparability between cells is demonstrated here, the author should consider rephrasing the claim. Also, what is relative expression?*

Response 8. Please note that the biaxial plots of CD3/CD3D versus CD19/CD19 in Figs 5H and 6E (6F in the revised manuscript), ITGA/ITGA6 versus EPCAM/EPCAM in Fig EV4H (Fig S9H in the revised manuscript), and BABAM1 versus LMNA in Fig EV4I (Fig 7A in the revised manuscript) indicate the recovery of the comparability between genes. Please also see **Responses 2 and 5**, and **Response 2** to Reviewer #1.

The "relative expression" means the expression level when the maximum log₂-transformed expression value in the data is set as 1. We provided an explanation of this point in the legends to Figs 5G, 6E, and EV4G (S9G in the revised manuscript) in the revised manuscript.

5. *The fairness of the comparison in Fig 6A remains unclear to me. Other methods do not include a prior dimension reduction step like PCA, but RECODE acts partly like a PCA dimension reduction. However all methods should have their expected up/downstream steps for best performance. How do other methods perform if a standard PCA step is placed between preprocessing and UMAP?*

Response 9. This comment is based on a misunderstanding of RECODE by the Reviewer. Please see **Response 3**.

Minor points

6. *The methods of RECODE have symbol errors and mixups, such as in the definition of t_j . The authors should double check all the symbols and equations in methods.*

Response 10

We would like to thank the Reviewers for pointing out this error, which we have corrected in the revised manuscript. We have double-checked the notations and definitions and have not been able to find any other essential symbol errors or mixups in the revised manuscript.

7. *The classification of genes is actually derived from the assumption $\text{Var}_i(x_{ij}^{\text{true}}) \ll \text{Var}_i(x_{ij})$. However only x_{ij}^{true} is constant is mentioned, and the relation between gene classification and this assumption is not clear in the paper. Clarifying these two in methods would make RECODE's applicability more understandable.*

Response 11. We have not assumed “ $\text{Var}_i(x_{ij}^{\text{true}}) \ll \text{Var}_i(x_{ij})$ ” for the gene classification. The gene classification is introduced to explain the applicability of RECODE, and we assume nothing in the definition of gene classification. The assumption of the RECODE applicability is only “that most genes are not directly related to cell identifications” (“Applicability of RECODE” in the **Methods and Materials**), and we have clearly mentioned the relation between this assumption and our settings in the same section.

8. *Related to point 1. "...the current scRNA-seq data analysis may not capture the true gene-to-gene relations..." I suppose the authors meant scRNA-seq data analysis without preprocessing?*

Response 12. The reviewer is correct; we were referring to the data without preprocessing. We clarified this in the revised manuscript (2nd paragraph in the “**Resolution of CODs in scRNA-seq data analysis by RECODE**” section).

Reviewer #3:

The authors have carefully considered feedback received and done major improvements in the clarity of explaining their method and in inclusion of additional experiments and benchmarks to support the validity of conclusions drawn.

The revised manuscript clearly demonstrates caveats in current best practices. The evaluation with simulated and real data is supporting key conclusions drawn regarding benefits of the proposed RECODE method against existing solutions. The method and its implementation is shared with the community. This work will contribute significantly conceptually and in the methodology available to analyze the single cell genomics that is

now widely adopted in the field of biomedicine.

Response 1. We would like to sincerely thank the Reviewer for his/her encouraging comments on our manuscript.

Minor comments:

The current Discussion is focused on additional elaboration of presented results and future directions but limited in placing the work in context of previous related work, or bringing up possible unexpected findings.

-The distinctions as made in figure panels on the types of approaches proposed in the field could be re-stated to summarize the context also in this section.

Response 2. In response to the Reviewer's comment, we have provided a discussion on the classification of the imputation methods and the approaches used in the RECODE in the 5th paragraph of the **Discussion** section in the revised manuscript.

-The suitability to 10X data is supported by data presented. Was it surprising that Smart-seq protocols gave different results?

Response 3. We would like to thank the Reviewer for this constructive comment. Smart-seq2 does not use UMIs, and therefore, the noise derived from PCR amplification is not canceled. On the other hand, Smart-seq3 involves UMI in the cDNA synthesis step, but was evaluated as weakly applicable to RECODE (Fig 4). In contrast to typical methods—such as scRNA-seq on the 10X chromium platform—that incorporate UMIs in the 3' ends of mRNAs at the beginning of cDNA synthesis, the Smart-seq3 method provides UMIs at the 5' ends of full-length cDNAs after the cDNA synthesis (Hagemann-Jensen et al., 2020). This would create additional noise, since the cDNA synthesis reaction often stops in the middle of mRNAs and full-length cDNA synthesis depends on various variables, such as mRNA length and nucleotide compositions (Nakamura et al., 2015). We reason that such (potential) sources of additional noise make Smart-seq2 and Smart-seq3 only weakly applicable to RECODE. We have provided a relevant discussion in the revised manuscript (2nd paragraph of the "**Discussion**" section).

-Batch effects: this is left for future work - however, would their method remedy some aspects of the challenge e.g would processing each sample within a batch with RECODE, followed by inclusion of existing batch normalization methods or inclusion of batch in DEG statistical models downstream represent a viable practical solution?

Response 4. Indeed, we are currently aiming to develop a method to eliminate the technical noise and batch effect simultaneously by combining RECODE with an existing batch normalization method, which we would like to present in a separate manuscript.

References

- Cao, Y., Yang, P., and Yang, J.Y.H. (2021). A benchmark study of simulation methods for single-cell RNA sequencing data. *Nat Commun* 12, 6911. 10.1038/s41467-021-27130-w.
- Chen, D., Sun, N., Hou, L., Kim, R., Faith, J., Aslanyan, M., Tao, Y., Zheng, Y., Fu, J., Liu, W., et al. (2019). Human Primordial Germ Cells Are Specified from Lineage-Primed Progenitors. *Cell Rep* 29, 4568-4582 e4565. 10.1016/j.celrep.2019.11.083.
- Hagemann-Jensen, M., Ziegenhain, C., Chen, P., Ramsköld, D., Hendriks, G.-J., Larsson, A.J.M., Faridani, O.R., and Sandberg, R. (2020). Single-cell RNA counting at allele and isoform resolution using Smart-seq3. *Nature Biotechnology* 38, 708-714. 10.1038/s41587-020-0497-0.
- Hao, Y., Hao, S., Andersen-Nissen, E., Mauck, W.M., Zheng, S., Butler, A., Lee, M.J., Wilk, A.J., Darby, C., Zager, M., et al. (2021). Integrated analysis of multimodal single-cell data. *Cell* 184, 3573-3587.e3529. 10.1016/j.cell.2021.04.048.
- Lähnemann, D., Köster, J., Szczurek, E., McCarthy, D.J., Hicks, S.C., Robinson, M.D., Vallejos, C.A., Campbell, K.R., Beerenwinkel, N., Mahfouz, A., et al. (2020). Eleven grand challenges in single-cell data science. *Genome Biology* 21. 10.1186/s13059-020-1926-6.
- Nakamura, T., Yabuta, Y., Okamoto, I., Aramaki, S., Yokobayashi, S., Kurimoto, K., Sekiguchi, K., Nakagawa, M., Yamamoto, T., and Saitou, M. (2015). SC3-seq: a method for highly parallel and quantitative measurement of single-cell gene expression. *Nucleic Acids Res* 43, e60. 10.1093/nar/gkv134.
- Pijuan-Sala, B., Griffiths, J.A., Guibentif, C., Hiscock, T.W., Jawaid, W., Calero-Nieto, F.J., Mulas, C., Ibarra-Soria, X., Tyser, R.C.V., Ho, D.L.L., et al. (2019). A single-cell molecular map of mouse gastrulation and early organogenesis. *Nature* 566, 490-495. 10.1038/s41586-019-0933-9.
- Satija, R., Farrell, J.A., Gennert, D., Schier, A.F., and Regev, A. (2015). Spatial reconstruction of single-cell gene expression data. *Nature Biotechnology* 33, 495-502. 10.1038/nbt.3192.
- Shah, S., Lubeck, E., Schwarzkopf, M., He, T.-F., Greenbaum, A., Sohn, C.H., Lignell, A., Choi, H.M.T., Gradinaru, V., Pierce, N.A., and Cai, L. (2016). Single-molecule RNA detection at depth via hybridization chain reaction and tissue hydrogel embedding and clearing. *Development* 143, 2862-2867. 10.1242/dev.138560.
- Tian, L., Dong, X., Freytag, S., Le Cao, K.A., Su, S., JalalAbadi, A., Amann-Zalcenstein, D., Weber, T.S., Seidi, A., Jabbari, J.S., et al. (2019). Benchmarking single cell RNA-sequencing analysis pipelines using mixture control experiments. *Nat Methods* 16, 479-487. 10.1038/s41592-019-0425-8.
- Torre, E., Dueck, H., Shaffer, S., Gospocic, J., Gupte, R., Bonasio, R., Kim, J., Murray, J., and Raj, A. (2018). Rare Cell Detection by Single-Cell RNA Sequencing as Guided by Single-Molecule RNA FISH. *Cell Syst* 6, 171-179 e175. 10.1016/j.cels.2018.01.014.

July 13, 2022

RE: Life Science Alliance Manuscript #LSA-2022-01591-TR

Prof. Mitinori Saitou
Institute for the Advanced Study of Human Biology, Kyoto University
Yoshida-Konoe-cho, Sakyo-ku
Kyoto 606-8501
Japan

Dear Dr. Saitou,

Thank you for submitting your revised manuscript entitled "Resolution of the curse of dimensionality in single-cell RNA sequencing data analysis". We would be happy to publish your paper in Life Science Alliance pending final revisions necessary to meet our formatting guidelines.

-please add ORCID ID for secondary corresponding author - you should have received instructions on how to do so
-please consult our manuscript preparation guidelines <https://www.life-science-alliance.org/manuscript-prep> and make sure your manuscript sections are in the correct order

A. FINAL FILES:

B. MANUSCRIPT ORGANIZATION AND FORMATTING:

Sincerely,

July 18, 2022

RE: Life Science Alliance Manuscript #LSA-2022-01591-TRR

Prof. Mitinori Saitou
Institute for the Advanced Study of Human Biology, Kyoto University
Yoshida-Konoe-cho, Sakyo-ku
Kyoto 606-8501
Japan

Dear Dr. Saitou,

Thank you for submitting your Research Article entitled "Resolution of the curse of dimensionality in single-cell RNA sequencing data analysis". It is a pleasure to let you know that your manuscript is now accepted for publication in Life Science Alliance. Congratulations on this interesting work.

DISTRIBUTION OF MATERIALS:

Again, congratulations on a very nice paper. I hope you found the review process to be constructive and are pleased with how the manuscript was handled editorially. We look forward to future exciting submissions from your lab.

Sincerely,
